# KW-DESIGN: PUSHING THE LIMIT OF PROTEIN DESIGN VIA KNOWLEDGE REFINEMENT

**Zhangyang Gao** [1,2,†], **Cheng Tan**[1,2,†], **Xingran Chen** [3], **Yijie Zhang** [4], **Jun Xia** [2], **Siyuan Li** [2]
**Stan Z. Li** [2,*]
[1] Zhejiang University; [2] Research Center for Industries of the Future, Westlake University
[3] University of Michigan, USA; [4] McGill University, Canada
{gaozhangyang, tancheng, Stan.ZQ.Li}@westlake.edu.cn

## ABSTRACT

Recent studies have shown competitive performance in protein inverse folding, while most of them disregard the importance of predictive confidence, fail to cover the vast protein space, and do not incorporate common protein knowledge. Given the great success of pretrained models on diverse protein-related tasks and the fact that recovery is highly correlated with confidence, we wonder whether this knowledge can push the limits of protein design further. As a solution, we propose a knowledge-aware module that refines low-quality residues. We also introduce a memory-retrieval mechanism to save more than 50% of the training time. We extensively evaluate our proposed method on the CATH, TS50, TS500, and PDB datasets and our results show that our KW-Design method outperforms the previous PiFold method by approximately 9% on the CATH dataset. KW-Design is the first method that achieves 60+% recovery on all these benchmarks. We also provide additional analysis to demonstrate the effectiveness of our proposed method. The code is publicly available via GitHub.

## 1 INTRODUCTION

Protein sequences, which are linear chains of amino acids, play a crucial role in determining the structure and function of cells and organisms. In recent years, there has been significant interest in designing protein sequences that can fold into desired structures (Pabo, 1983). Deep learning models (Li et al., 2014; Wu et al., 2021; Pearce & Zhang, 2021; Ovchinnikov & Huang, 2021; Ding et al., 2022; Gao et al., 2020; 2022a; Dauparas et al., 2022; Ingraham et al., 2019; Jing et al., 2020; Tan et al., 2022c; Hsu et al., 2022; O'Connell et al., 2018; Wang et al., 2018; Qi & Zhang, 2020; Strokach et al., 2020; Chen et al., 2019; Zhang et al., 2020a; Huang et al., 2017; Anand et al., 2022; Strokach & Kim, 2022; Li & Koehl, 2014; Greener et al., 2018; Karimi et al., 2020; Anishchenko et al., 2021; Cao et al., 2021; Liu et al., 2022; McPartlon et al., 2022; Huang et al., 2022; Dumortier et al., 2022; Li et al., 2022a; Maguire et al., 2021; Li et al., 2022b) have made significant progress in this area. However, many of these methods either ignore the importance of predictive confidence, fail to cover the vast protein space, or lack consideration of common protein knowledge. We argue that the absence of common protein knowledge limits the generalizability of protein design models, and propose a confidence-aware module that refines low-quality residues using pretrained features.

Previous protein design methods have not fully utilized predictive confidence, which is the maximum probability of a residue. Using PiFold (Gao et al., 2023) as our baseline, a recent competitive protein design model, we observed significant differences in confidence distributions between positive and negative residues, as shown in Fig. 1. This finding inspired us to propose a confidence-aware module that automatically identifies low-quality residues and iteratively refines them to reduce prediction errors. However, we encountered a challenge: the recovery rate of PiFold plateaued at around 52% regardless of how many refining layers (PiGNNs) were added. We hypothesized that this was due to the model being trapped in a local optimum based on the current training set. Escaping the local minimum would require additional inductive bias from other teacher models.

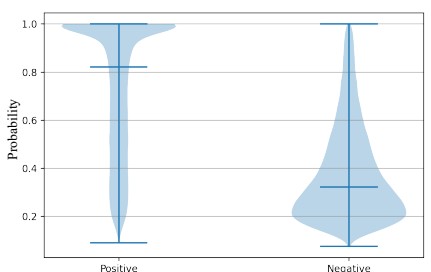

**Figure 1:** Confidence distributions of positive and negative residues designed by Pi-FoldGao et al. (2023). Positive residues are identical to native residues and vice versa.

---

†Equal Contribution, *Corresponding Author.

To escape the local minimum and improve the performance of our protein design model, we suggest leveraging pretrained teacher models. These models have made significant progress on a variety of downstream tasks (Zhang et al., 2022b; Meier et al., 2021; Zhang et al., 2022a; Chen et al., 2023) by learning common knowledge across a vast protein space. The structural knowledge (Zhang et al., 2022b; Hsu et al., 2022) can help to learn expressive protein features, while the sequential knowledge (Meier et al., 2021) can aid in designing rational proteins. In this study, we investigate three pretrained models, i.e., ESM (Meier et al., 2021; Lin et al., 2022), ESM-IF (Hsu et al., 2022), and GearNet (Zhang et al., 2022b), to extract structural and sequential embeddings as prior knowledge that can enhance our refinement module. As a structure-in and sequence-out task, the structural-based protein design can benefit from the multimodal knowledge and automatically revise residues that violate common sense.

To boost protein design, we propose a confidence-aware refining model that leverages multimodal knowledge. However, we face several challenges: (1) how to adaptively fuse multimodal pretrained knowledge based on predictive confidence, (2) how to develop more effective refining technologies, and (3) how to efficiently tune the model with large-scale pretrained parameters. Firstly, we propose a multimodal fusion module that combines the knowledge of structure pretraining, sequence pretraining, and history predictions. Predictive confidence is used to control the combination through gated attention, enabling the model to adaptively select proper knowledge. Secondly, we suggest using virtual MSA and recycling technologies to improve the recovery. Thirdly, we introduce a memory-retrieval mechanism that caches the intermediate results of modules. This mechanism enables the model to retrieve historical embeddings without performing a forward pass, resulting in more than 50% training time savings.

We call our method KW-Design, a refining method that considers multimodal knowledge as well as predictive confidence. We evaluate our method on four benchmark datasets: CATH, TS50, TS500 and PDB, and observe significant improvements across all settings. For example, KW-Design is the first method to achieve 60+% recovery on all these datasets. On the CATH dataset, we observe 9.11% improvement compared to the previous PiFold method. We also conduct extensive ablation studies to demonstrate how the knowledge-refinement module works and how the memory-retrieval mechanism saves training time. Additionally, we provide further analysis to demonstrate the superiority of our proposed method. Overall, our results demonstrate the effectiveness of our approach in improving the recovery of protein sequence design. In the appendix, we show that the recovery is highly correlated with the structural similarity and discuss that under the same level of structural similarity, our method can generate more novel protein sequences than baselines.

## 2 RELATED WORK

Recently, AI algorithms have evolved rapidly in many fields (Gao et al., 2022d; Cao et al., 2022; Tan et al., 2022a; Li et al., 2022c; He et al., 2020; Stärk et al., 2022), where the protein folding problem (Jumper et al., 2021; Wu et al., 2022; Lin et al., 2022; Mirdita et al., 2022; Wang et al., 2022; Li et al., 2022d) that has troubled humans for decades has been nearly solved. Its inverse problem-structure-based protein design - is receiving increasing attention.

**Problem definition** The structure-based protein design aims to find the amino acids sequence $\mathcal{S} = \{s_i : 1 \leq i \leq n\}$ folding into the desired structure $\mathcal{X} = \{\boldsymbol{x}_i \in \mathbb{R}^3 : 1 \leq i \leq n\}$, where $n$ is the number of residues and the natural proteins are composed by 20 types of amino acids, i.e., $1 \leq s_i \leq 20$ and $s_i \in \mathbb{N}^+$. Formally, that is to learn a function $\mathcal{F}_\theta$:

$$\mathcal{F}_\theta : \mathcal{X} \mapsto \mathcal{S}. \tag{1}$$

Because homologous proteins always share similar structures (Pearson & Sierk, 2005), the problem itself is underdetermined, i.e., the valid amino acid sequence may not be unique (Gao et al., 2020).

**MLP-based models** MLP is used to predict the probability of 20 amino acids for each residue, and various methods are mainly difficult in feature construction. These methods are commonly evaluated on the TS50, which contains 50 native structures. For example, SPIN (Li et al., 2014) achieves 30% recovery on TS50 by using torsion angles ($\phi$ and $\psi$), sequence profiles, and energy profiles. Through adding backbone angles ($\theta$ and $\tau$), local contact number, and neighborhood distance, SPIN2 (O'Connell et al., 2018) improves the recovery to 34%. Wang's model (Wang et al.,

2018) suggests using backbone dihedrals ($\phi$, $\psi$ and $\omega$), the solvent accessible surface area of backbone atoms ($C_\alpha$, $N$, $C$, and $O$), secondary structure types (helix, sheet, loop), $C_\alpha - C_\alpha$ distance and unit direction vectors of $C_\alpha - C_\alpha$, $C_\alpha - N$ and $C_\alpha - C$ and achieves 33% recovery. The MLP method enjoys a high inference speed but suffers from a low recovery rate because the structural information is not sufficiently considered.

**CNN-based models**   These methods use 2D CNN or 3D CNN to extract protein features (Torng & Altman, 2017; Boomsma & Frellsen, 2017; Weiler et al., 2018; Zhang et al., 2020a; Qi & Zhang, 2020; Chen et al., 2019) and are commonly evaluated on the TS50 and TS500. SPROF (Chen et al., 2019) adopts 2D CNN to learn residue representations from the distance matrix and achieves a 40.25% recovery on TS500. 3D CNN-based methods, such as ProDCoNN (Zhang et al., 2020a) and DenseCPD (Qi & Zhang, 2020), extract residue features from the atom distribution in a three-dimensional grid box. For each residue, after being translated and rotated to a standard position, the atomic distribution is fed to the model to learn translation- and rotation-invariant features. ProD-CoNN (Zhang et al., 2020a) designs a nine-layer 3D CNN with multi-scale convolution kernels and achieves 42.2% recovery on TS500. DenseCPD (Qi & Zhang, 2020) uses the DensetNet architecture (Huang et al., 2017) to boost the recovery to 55.53% on TS500. Recent works (Anand et al., 2022) have also explored the potential of deep models to generalize to *de novo* proteins. Despite the improved recovery achieved by the 3D CNN models, their inference is slow, probably because they require separate preprocessing and prediction for each residue.

**Graph-based models**   These methods use $k$-NN graph to represent the 3D structure and employ graph neural networks (Defferrard et al., 2016; Kipf & Welling, 2016; Veličković et al., 2017; Zhou et al., 2020; Zhang et al., 2020b; Gao et al., 2022b; Tan et al., 2022b; Gao et al., 2022c) to extract residue features while considering structural constraints. The protein graph encodes residue information and pairwise interactions as the node and edge features, respectively. GraphTrans (Ingraham et al., 2019) uses the graph attention encoder and autoregressive decoder for protein design. GVP (Jing et al., 2020) proposes geometric vector perceptrons to learn from both scalar and vector features. GCA (Tan et al., 2022c) introduces global graph attention for learning contextual features. In addition, ProteinSolver (Strokach et al., 2020) is developed for scenarios where partial sequences are known while not reporting results on standard benchmarks. Recently, AlphaDesign (Gao et al., 2022a), ProteinMPNN (Dauparas et al., 2022) and Inverse Folding (Hsu et al., 2022) achieve dramatic improvements. Compared to CNN methods, graph models do not require rotating each residue separately as in CNN, thus improving the training efficiency. Compared to MLP methods, the well-exploited structural information helps GNN obtain higher recovery.

# 3   METHOD

## 3.1   OVERALL FRAMEWORK

The framework of our KW-Design model is illustrated in Figure 2, comprising an initial design model ($F_{\theta^{(0)}}$), and $L$ confidence-aware knowledge-tuning modules ($f_{\phi^{(1)}}, f_{\phi^{(2)}}, \cdots, f_{\phi^{(L)}}$), where $\theta^{(0)}$ and $\phi^{(i)}$ are learnable parameters. Each knowledge tuning process consists of a knowledge extractor (pretrained ESMIF, ESM2-650M, or GearNet), a confidence predictor, a multimodal fusion layer, and a refinement module. Note that pretrained protein models are frozen during optimizing and not included in $\phi^{(i)}$. To simplify the notation, we write $F_{\theta^{(k)}} = f_{\phi^{(k)}} \cdots \circ f_{\phi^{(1)}} \circ F_{\theta^{(0)}}$, where $\theta^{(k)} = \phi^{(k)} || \cdots || \phi^{(2)} || \phi^{(1)} || \theta^{(0)}$, and $||$ denotes concatenation operation.

For the $l$-th knowledge-tuning module, we denote the protein structure as $\boldsymbol{x} \in \mathbb{R}^{n,3}$, the residue embedding as $\boldsymbol{h}^{(l)} \in \mathbb{R}^{n,d}$, and the predicted probabilities as $\boldsymbol{p}^{(l)} \in \mathbb{R}^{n,21}$. Formally, we have $\boldsymbol{h}^{(l)} = f_{\phi^{(l)}} \cdots \circ f_{\phi^{(1)}} \circ F_{\theta^{(0)}}(\boldsymbol{x})$ and $\boldsymbol{p}^{(l)} = \texttt{Predict}^{(l)}(\boldsymbol{h}^{(l)})$. Here, $\circ$ denotes the operation of compositing functions, $n$ is the number of residues, $d$ is the embedding size, 21 is the number of amino acids plus a special token of [mask], and $\texttt{Predict}^{(l)}(\cdot | T)$ is a linear layer followed by the softmax activation using a temperature value of $T$, i.e., $\texttt{Softmax}(\texttt{Linear}(\cdot)/T)$. The overall objective of our KW-Design model is to minimize the loss function $\mathcal{L}$ with respect to the learnable parameters $\theta^{(0)}, \phi^{(1)}, \cdots, \phi^{(L)}$:

$$\min_{\theta^{(0)}, \phi^{(1)}, \cdots, \phi^{(L)}} \mathcal{L}(f_{\phi^{(L)}} \cdots \circ f_{\phi^{(1)}} \circ F_{\theta^{(0)}}(\boldsymbol{x}), \boldsymbol{s}) \tag{2}$$

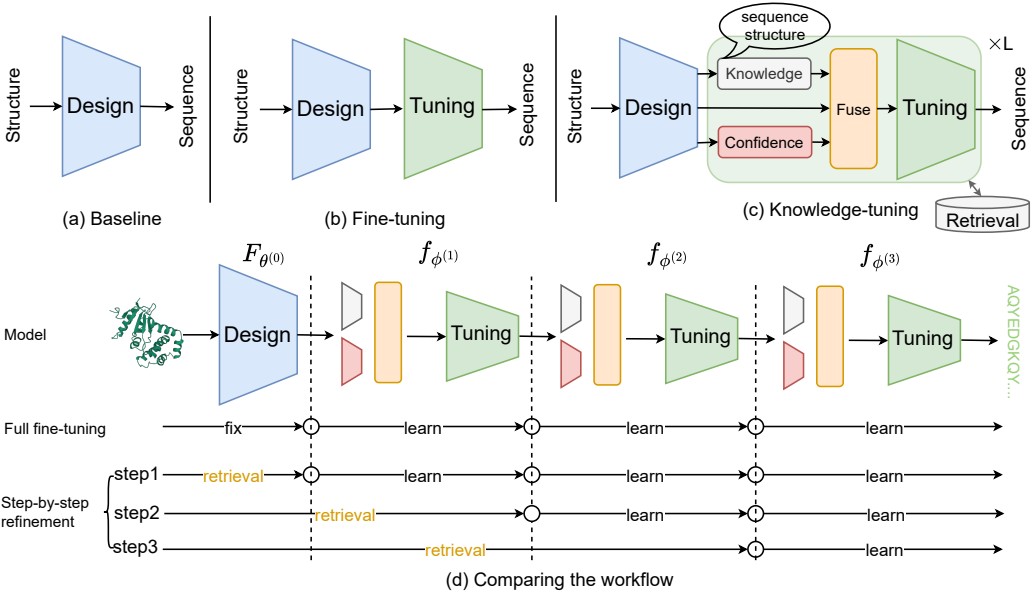

**Figure 2:** Comparison of various models. (a) The baseline trains from scratch without using pretrained knowledge. (b) The fine-tuning model refines the output of the baseline. (c) The knowledge-tuning model fuses multimodal pretained knowledge and the confidence to enhance the refinement module. (d) The proposed KW-Design model introduces a memory bank to speed up the training process by skipping the forward pass of well-tuned modules.

Here, $s$ is the reference sequence, and $x$ is the protein structure.

Under the assumptions that if $\mathcal{L}(F_{\theta^{(k)}}(x), s) < \mathcal{L}(F_{\theta^{(k)'}}, s)$, then $\mathcal{L}(f_{\phi^{(k+1)}}(F_{\theta^{(k)}}(x)), s) < \mathcal{L}(f_{\phi^{(k+1)}}(F_{\theta^{(k)'}}(x)), s)$, indicating that a stronger design model leads to better results with the same fine-tuning module, we simplify the objective as:

$$\min_{\phi^{(k)}} \mathcal{L}(f_{\phi^{(k)}}(h^{(k-1)}), s), s.t., \theta^{(k-1)} = \min_{\theta^{(k-1)}} \mathcal{L}(F_{\theta^{(k-1)}}(x), s) \qquad (3)$$

Note that $h^{(k-1)} = F_{\theta^{(k-1)}}(x), k \in \{L, L-1, \cdots, 1\}$. Eq.(3) suggests the problem could be solved by optimizing the fine-tuning modules $f_{\phi^{(1)}}, f_{\phi^{(2)}}, \cdots, f_{\phi^{(L)}}$ sequentially. Therefore, we conduct module-wise training strategy, where the parameters $\theta^{(k-1)}$ are frozen when optimizing $\phi^{(k)}$. In the appendix A, we show the module-wise training strategy is helpful in reducing the computational cost by avoiding redundant forward passes of large pretrained models. The optimal embedding in the memory bank is automatically determined by the early stopping (patience=5), where the indicator is the average predictive confidence of the whole sequence. We will introduce the details of the refining technique, knowledge-tuning module and memory bank in the following sections.

## 3.2 REFINING TECHNIQUE

**Recycling Process**   Given the initial residue embedding $h^{(0)} = F_{\theta^{(0)}}(x)$, our KW-Design applies a sequence of knowledge-tuning modules to update the residue embedding:

$$h^{(0)} \xrightarrow{f_{\phi^{(1)}}} \cdots h^{(l)} \xrightarrow{f_{\phi^{(l+1)}}} h^{(l+1)} \xrightarrow{f_{\phi^{(l+1)}}} h^{(l+1)} \cdots \xrightarrow{f_{\phi^{(L)}}} h^{(L)} \qquad (4)$$

where $L$ is the maximum number of refinement modules. The predictive probability is obtained by:

$$p^{(l)} = \texttt{Predict}^{(l)}(h^{(l)}|T) \qquad (5)$$

**Virtual MSA**   To capture diverse protein knowledge, we sample a set of protein sequences $\{s^{(l),i} \sim \text{Multinomial}(p^{(l)})|0 \le i < m\}$ from the predicted probabilities by setting $T = 0.05$ in Eq. 5. Any residues with a confidence score below 0.99 are replaced with $\texttt{[mask]}$, which helps in constructing a high-quality prompt template. We call this set of sequences the virtual multiple

sequence alignment (MSA) and feed them into pretrained models to obtain the residue embeddings:

$$\boldsymbol{h}_{seq}^{(l),i} = \mathcal{F}_{seq}(\boldsymbol{s}^{(l),i}) \tag{6}$$

$$\boldsymbol{h}_{3d}^{(l),i} = \mathcal{F}_{3d}(\boldsymbol{s}^{(l),i}, \boldsymbol{x}) \tag{7}$$

where $\boldsymbol{x}$ is the 3d coordinates of residues, and $\mathcal{F}_{seq}$ and $\mathcal{F}_{3d}$ are sequence and structure pretrained models, respectively. The sequential embedding $\boldsymbol{h}_{seq}^{(l),i} \in \mathbb{R}^{n,d_{seq}}$ captures the primary sequence knowledge using ESM2-650M, while the structural embedding $\boldsymbol{h}_{3d}^{(l),i} \in \mathbb{R}^{n,d_{3d}}$ captures the structure knowledge using ESMIF or GearNet. Together, these features are combined as a unified embedding $\boldsymbol{z}^{(l)}$ through the fusion module:

$$\boldsymbol{z}^{(l)} = \texttt{Fuse}(\{\boldsymbol{h}_{seq}^{(l),i}\}_{i=1}^m, \{\boldsymbol{h}_{3d}^{(l),i}\}_{i=1}^m, \{\boldsymbol{s}^{(l),i}\}_{i=1}^m, \boldsymbol{p}^{(l)}) \tag{8}$$

which can be further converted as $\boldsymbol{h}^{(l+1)}$ and $\boldsymbol{p}^{(l+1)}$:

$$\boldsymbol{h}^{(l+1)} = \texttt{Refine}^{(l+1)}(\boldsymbol{z}^{(l)}); \boldsymbol{p}^{(l+1)} = \texttt{Predict}^{(l+1)}(\boldsymbol{h}^{(l+1)}|T) \tag{9}$$

where the $\texttt{Refine}^{(l+1)}$ module is the 10-layer PiGNNs, described later in Section 3.3.

**Confidence-aware updating** We define the confidence vector of a sequence $\boldsymbol{s}$ as the corresponding predictive probability, written as $\boldsymbol{c_s}$:

$$\boldsymbol{c_s} = [p_{1,s_1}, p_{2,s_2}, \cdots, p_{n,s_n}]^T \tag{10}$$

Note that $p_{i,s_i}$ represents the predicted probability of the $i$-th amino acid and the predicted residue type is $s_i$. Because some residues are harder to design than others, they may benefit more from refinement. Considering this, we introduce a confidence-aware gated attention mechanism that updates the pre- and post-refinement embeddings based on the predictive confidence of each residue. This allows us to focus more on difficult residues during refinement and improve overall design performance:

$$\boldsymbol{h}^{(l+1)} \leftarrow \boldsymbol{h}^{(l+1)} \odot \sigma(\text{MLP}_1(\boldsymbol{c}_{\boldsymbol{s}'}^{(l+1)} - \boldsymbol{c}_{\boldsymbol{s}}^{(l)})) + \boldsymbol{h}^{(l)} \odot \sigma(\text{MLP}_2(\boldsymbol{c}_{\boldsymbol{s}}^{(l)} - \boldsymbol{c}_{\boldsymbol{s}'}^{(l+1)})) \tag{11}$$

where $\boldsymbol{s} \sim \text{Multinomial}(\boldsymbol{p}^{(l)})$ and $\boldsymbol{s}' \sim \text{Multinomial}(\boldsymbol{p}^{(l+1)})$ are sampled from multimodal distributions. $\sigma$ is the sigmoid function, $\odot$ is element-wise multiplication.

## 3.3 KNOWLEDGE-TUNING MODULE

The knowledge-tuning module updates the residue embeddings of well-tuned models to generate more rational protein sequences. As shown in Figure 2, the knowledge-tuning module includes a knowledge extractor, a confidence predictor, a fusion layer, and a tuning layer.

**Knowledge extractor & Confidence predictor** As introduced in Sec.3.2, the pretrained knowledge extractors ($\mathcal{F}_{seq}^{(l)}$ and $\mathcal{F}_{3d}^{(l)}$) extract embeddings from virtual MSAs. The confidence predictor ($\texttt{Predict}^{(l+1)}$) outputs the predictive probability $\boldsymbol{p}^{(l+1)}$ based on the residue embedding $\boldsymbol{h}^{(l+1)}$.

**Fusion layer** The fusion layer combines the sequential and structural embeddings with a confidence score to obtain a unified embedding. Specifically, the structural and sequential MSA embeddings are fused using a confidence-aware gated layer:

$$\boldsymbol{z}^{(l)} = \sum_{i=1}^m \left[ \text{Embed}(\boldsymbol{s}^{(l),i}) + \text{MLP}_3(\boldsymbol{h}_{seq}^{(l),i}) + \text{MLP}_4(\boldsymbol{h}_{3d}^{(l),i}) \right] \odot \sigma(\text{MLP}_5(\boldsymbol{c}_{\boldsymbol{s}}^{(l),i})) \tag{12}$$

**Refinement module** The refinement module is a learnable graph neural network (GNN) that takes $\boldsymbol{z}^{(l)}$ as input node features and $\boldsymbol{e}^{(0)}$ as input edge features. Given the protein graph $\mathcal{G}$, node features $\boldsymbol{z}^{(l)}$ and edge features $\boldsymbol{e}^{(l)}$, we use PiGNNs as the refinement module for updating embeddings:

$$\boldsymbol{h}^{(l+1)}, \boldsymbol{e}^{(l+1)} = \texttt{PiGNNs}(\boldsymbol{z}^{(l)}, \boldsymbol{e}^{(l)}, \mathcal{G}) \tag{13}$$

Note that the initial node features $\boldsymbol{z}^{(0)}$ and edge features $\boldsymbol{e}^{(0)}$ are extracted from PiFold.

### 3.4 MEMORY RETRIEVAL

From Eq.3, we know that $\theta^{(0)}, \phi^{(1)}, \cdots, \phi^{(l)}$ are frozen parameters when optimizing $\phi^{(l+1)}$. Therefore, we can use a memory bank $\mathcal{M}^{(l)}$ to store and retrieve the intermediate embeddings of the $l$-th design model $F_{\theta^{(l)}}(\boldsymbol{x})$ for speeding up the process of optimizing $\phi^{(l+1)}$. As shown in Alg.1, the protein embedding $\boldsymbol{h}^{(l)}$ can be retrieved from the memory bank $\mathcal{M}^{(l)}$ without the need for a forward pass, provided that the following conditions are satisfied: (1) the embedding $\boldsymbol{h}^{(l)}$ is already stored in $\mathcal{M}^{(l)}$ and (2) the saved embeddings are consistently obtained from an optimal model $F_{\theta^{(l)}}$. While the first condition is straightforward, the second condition requires the algorithm to automatically determine the optimal $\phi^{(l)}$ and freeze $f_{\phi^{(l)}}$ to ensure that the memorized embeddings are consistent. To determine the optimal $\phi^{(l)}$, we use the average confidence score over the validation set as an indicator and apply the early stopping operation to determine the optimal $\phi^{(l)}$, with the patience of 5.

---

**Algorithm 1** Memory Net Framework

**Usage**: Retrieve embedding from the memory bank without the forward pass.

---

**Input:** A batch of inputs $\mathcal{H}_{1:b}^{(l)} = \boldsymbol{h}_1^{(l)}||\boldsymbol{h}_2^{(l)}||\cdots||\boldsymbol{h}_b^{(l)}$;

**Output:** A batch of outputs $\mathcal{H}_{1:b}^{(l+1)} = \boldsymbol{h}_1^{(l+1)}||\boldsymbol{h}_2^{(l+1)}||\cdots||\boldsymbol{h}_b^{(l+1)}$.

1: Step1: Debatch input data
2: $\{\boldsymbol{h}_1^{(l)}, \boldsymbol{h}_2^{(l)}, \cdots, \boldsymbol{h}_b^{(l)}\} = \text{DeBatch}(\mathcal{H}_{1:b}^{(l)})$;
3:
4: Step2: Retrieve embeddings
5: **for** $i \in [0, b)$
6:     **if** $\boldsymbol{h}_i^{(l+1)} \in \mathcal{M}^{(l+1)}$ and $f_{\phi^{(l)}}$ is early stopped
7:         $\boldsymbol{h}_i^{(l+1)} = \mathcal{M}^{(l+1)}[i]$
8:     **else**
9:         $\boldsymbol{h}_i^{(l+1)} = \text{Refine}^{(l+1)}(\boldsymbol{h}_i^{(l)})$
10:
11: Step3: Batch output
12: Save $\{\boldsymbol{h}_1^{(l+1)}, \boldsymbol{h}_2^{(l+1)}, \cdots, \boldsymbol{h}_b^{(l+1)}\}$ to $\mathcal{M}^{(l+1)}$
13: Return $\boldsymbol{h}_1^{(l+1)}||\boldsymbol{h}_2^{(l+1)}||\cdots||\boldsymbol{h}_b^{(l+1)}$

---

## 4 EXPERIMENTS

We evaluate the performance of KW-Design on multiple datasets, including CATH4.2, CATH4.3, TS50, TS500, and PDB. We also conduct systematic studies to answer the following questions:

- **Performance (Q1):** Can KW-Design achieve state-of-the-art accuracy on real-world datasets?
- **Refining technology (Q2):** How much can models gain from different refinement techniques?
- **Knowledge tuning (Q3):** Which pretrained knowledge is helpful in improving protein design, and how much of a speed boost can the memory bank bring?
- **More analysis (Q4):** How does the KW-Design make a difference on the basis of PiFold?

### 4.1 PERFORMANCE ON CATH (Q1)

**Objective & Setting** We demonstrate the effectiveness of KW-Design on the widely used CATH (Orengo et al., 1997) dataset. To provide a comprehensive comparison, we conduct experiments on both CATH4.2 and CATH4.3. The CATH4.2 dataset consists of 18,024 proteins for training, 608 proteins for validation, and 1,120 proteins for testing, following the same data splitting as Graph-Trans (Ingraham et al., 2019), GVP (Jing et al., 2020), and PiFold (Gao et al., 2023). The CATH4.3 dataset includes 16,153 structures for the training set, 1,457 for the validation set, and 1,797 for the test set, following the same data splitting as ESMIF (Hsu et al., 2022). The model is trained up to 20 epochs using the Adam optimizer on an NVIDIA V100. The batch size and learning rate used for training are 32 and 0.001, respectively. To evaluate the generative quality, we report perplexity and median recovery scores on short-chain, single-chain, and all-chain settings.

**Baselines** We compare KW-Design with recent graph models, including StructGNN, StructTrans (Ingraham et al., 2019), GCA (Tan et al., 2022c), GVP (Jing et al., 2020), GVP-large, AlphaDesign (Gao et al., 2022a), ESM-IF (Hsu et al., 2022), ProteinMPNN (Dauparas et al., 2022), and PiFold Gao et al. (2023), as most of them are open-source. To provide a head-to-head comparison with ESMIF, we retrain our model on the CATH4.3 dataset following the same data splitting as ESMIF, while we do not utilize AF2DB for training the model.

**Results & Analysis** From Table 1, we conclude that KW-Design consistently achieves state-of-the-art performance on different settings. Specifically, we observe the following: (1) KW-Design is the first model to exceed 60% recovery on both CATH4.2 and CATH4.3, demonstrating its superior ability in generating protein structures. (2) On the full CATH4.2 dataset, KW-Design achieves a perplexity of 3.46 and a recovery of 60.77%, outperforming the previous state-of-the-art model PiFold by 23.95% and 9.11%, respectively. Furthermore, KW-Design achieves a recovery improvement of

**Table 1:** Results comparison on the CATH dataset. All baselines are reproduced under the same code framework, except ones marked with †. We copy results of GVP-large and ESM-IF from their manuscripts (Hsu et al., 2022). The **best** and underline{suboptimal} results are labeled with bold and underline.

| Model | Perplexity ↓ | | | Recovery % ↑ | | | CATH version | |
|---|---|---|---|---|---|---|---|---|
| | Short | Single-chain | All | Short | Single-chain | All | 4.2 | 4.3 |
| StructGNN | 8.29 | 8.74 | 6.40 | 29.44 | 28.26 | 35.91 | ✓ | |
| GraphTrans | 8.39 | 8.83 | 6.63 | 28.14 | 28.46 | 35.82 | ✓ | |
| GCA | 7.09 | 7.49 | 6.05 | 32.62 | 31.10 | 37.64 | ✓ | |
| GVP | 7.23 | 7.84 | 5.36 | 30.60 | 28.95 | 39.47 | ✓ | |
| GVP-large† | 7.68 | _6.12_ | 6.17 | 32.6 | 39.4 | 39.2 | | ✓ |
| AlphaDesign | 7.32 | 7.63 | 6.30 | 34.16 | 32.66 | 41.31 | ✓ | |
| ESM-IF† | 8.18 | 6.33 | 6.44 | 31.3 | 38.5 | 38.3 | | ✓ |
| ESM-IF (AF2DB)† | 6.05 | 4.00 | 4.01 | 38.1 | **51.5** | 51.6 | | ✓ |
| ProteinMPNN | 6.21 | 6.68 | 4.61 | 36.35 | 34.43 | 45.96 | ✓ | |
| PiFold | _6.04_ | 6.31 | _4.55_ | _39.84_ | 38.53 | _51.66_ | ✓ | |
| KW-Design (Ours) | **5.48** | **5.16** | **3.46** | **44.66** | _45.45_ | **60.77** | ✓ | |
| KW-Design (Ours) | **5.47** | **5.23** | **3.49** | **43.86** | _45.95_ | **60.38** | | ✓ |

4.82% and 6.92% on the short and single-chain settings, respectively. (3) KW-Design also achieves similar improvements when extending to the CATH4.3 dataset, further validating its effectiveness and generalizability. Overall, the strong performance suggests that KW-Design could be a valuable tool for advancing protein engineering and drug design.

## 4.2 PERFORMANCE ON TS50 AND TS500 (Q1)

**Objective & Setting** To provide a more comprehensive evaluation and demonstrate the generalizability of KW-Design, we also evaluate it on two standard protein benchmarks, TS50 and TS500. These datasets contain 50 and 500 proteins, respectively, and are widely used for evaluation.

**Table 2:** Results on TS50 and TS500. All baselines are reproduced under the same code framework.

| Model | TS50 | | | TS500 | | |
|---|---|---|---|---|---|---|
| | Perplexity ↓ | Recovery ↑ | Worst ↑ | Perplexity ↓ | Recovery ↑ | Worst ↑ |
| StructGNN | 5.40 | 43.89 | 26.92 | 4.98 | 45.69 | _0.05_ |
| GraphTrans | 5.60 | 42.20 | 29.22 | 5.16 | 44.66 | 0.03 |
| GVP | 4.71 | 44.14 | 33.73 | 4.20 | 49.14 | **0.09** |
| GCA | 5.09 | 47.02 | 28.87 | 4.72 | 47.74 | 0.03 |
| AlphaDesign | 5.25 | 48.36 | 32.31 | 4.93 | 49.23 | 0.03 |
| ProteinMPNN | 3.93 | 54.43 | 37.24 | 3.53 | 58.08 | 0.03 |
| PiFold | _3.86_ | _58.72_ | _37.93_ | _3.44_ | _60.42_ | 0.03 |
| KW-Design (Ours) | **3.10** | **62.79** | **39.31** | **2.86** | **69.19** | 0.02 |

**Results & Analysis** Experimental results are shown in Table.2, where KW-Design significantly outperforms previous baselines on all benchmarks. We observe that: (1) On the TS50 dataset, KW-Design achieves a perplexity of 3.10 and a recovery rate of 62.79%, outperforming the previous state-of-the-art model, PiFold, by 19.69% and 4.07%. (2) On the TS500 dataset, KW-Design achieves a perplexity of 2.86 and a recovery rate of 69.19%, outperforming PiFold by 16.86% and 8.77%. (3) Notably, KW-Design is the first model to exceed 60% and 65% recovery on the TS50 and TS500 benchmarks, respectively.

## 4.3 PERFORMANCE ON PDB (Q1)

**Objective & Setting** In addition, we expand the evaluation to include multi-chain protein design using the dataset curated by ProteinMPNN. The dataset was preprocessed by clustering sequences at 30% identity, resulting in 25,361 clusters. Following ProteinMPNN's setup, we divided the clusters randomly into training (23,358), validation (1,464), and test sets (1,539), ensuring that none of the chains from the target chain or biounits of the target chain were present in the other two sets. During each training epoch, we cycled through the sequence clusters and randomly selected a sequence

member from each cluster. All baselines are retrained and evaluated under the PDB dataset. We report the median values of confidence scores and recovery rates for the designed sequences.

**Table 3:** Multi-chain results on the PDB dataset.

| Model length | Confidence ↑ | | | | Recovery % ↑ | | | |
|---|---|---|---|---|---|---|---|---|
| | $L < 100$ | $100 \leq L < 500$ | $500 \leq L < 1000$ | Full | $L < 100$ | $100 \leq L < 500$ | $500 \leq L < 1000$ | Full |
| StructGNN | 0.49 | 0.49 | 0.50 | 0.49 | 0.41 | 0.41 | 0.42 | 0.41 |
| GraphTrans | 0.48 | 0.47 | 0.48 | 0.48 | 0.40 | 0.39 | 0.40 | 0.40 |
| GCA | 0.45 | 0.45 | 0.46 | 0.45 | 0.41 | 0.41 | 0.42 | 0.41 |
| GVP | 0.51 | 0.53 | 0.55 | 0.54 | 0.44 | 0.42 | 0.45 | 0.43 |
| AlphaDesign | 0.52 | 0.53 | 0.54 | 0.53 | 0.48 | 0.49 | 0.50 | 0.49 |
| ProteinMPNN | 0.54 | 0.56 | 0.58 | 0.57 | 0.52 | 0.53 | 0.55 | 0.53 |
| PiFold | 0.56 | 0.60 | 0.63 | 0.61 | 0.54 | 0.58 | 0.60 | 0.58 |
| **KWDesign** | **0.65** | **0.71** | **0.74** | **0.71** | **0.59** | **0.66** | **0.67** | **0.66** |

**Results & Analysis** From Table 3, we conclude that: (1) The predictive confidence score, an unsupervised metric, exhibits a strong correlation with the recovery. As a result, we leverage this metric for two purposes: firstly, to fuse pre-trained knowledge, and secondly, to construct high-quality sequence prompts. (2) The longer the protein, the higher the recovery. Given that the protein length in the PDB dataset is generally larger than that in the CATH dataset, all models achieve higher recovery rates on the PDB dataset. (3) KW-Design achieves the best performance on all protein lengths, demonstrating its effectiveness in multi-chain protein design.

## 4.4 Refining technology (Q2)

**Objective & Setting** We conduct ablation studies to investigate the effects of virtual MSA, recycling, and the confidence-aware tuning module. We follow the same experimental setting as in Section 4.1 and report the results on the CATH dataset. Specifically, we vary the number of virtual MSA and recycling times from 1 to 3, and remove the confidence-aware tuning module by replacing the confidence score with a constant value of 1.0. We also compare the training time with and without using the memory bank.

| Config | | | Perplexity ↓ | | | Recovery % ↑ | | | Training time (per epoch) ↓ | |
|---|---|---|---|---|---|---|---|---|---|---|
| w/o confidence | MSA | Recycle | Short | Single-chain | All | Short | Single-chain | All | w/o memory | w memory |
| | 1 | 1 | 5.54 | 5.39 | 3.59 | 42.58 | 42.74 | 58.39 | 20min | 70 min |
| | 1 | 2 | 5.52 | 5.31 | 3.52 | 44.72 | 44.19 | 59.72 | 40min | 140min |
| | 1 | 3 | 5.46 | 5.17 | 3.48 | 43.91 | 44.16 | 60.34 | 60min | 210min |
| | 1 | 1 | 5.54 | 5.39 | 3.59 | 42.58 | 42.74 | 58.39 | 20min | 70 min |
| | 2 | 1 | 5.55 | 5.42 | 3.56 | 42.72 | 42.16 | 58.62 | 33min | 83min |
| | 3 | 1 | 5.57 | 5.42 | 3.56 | 42.94 | 43.23 | 58.71 | 45min | 95min |
| | 2 | 2 | 5.49 | 5.22 | 3.49 | 44.76 | 45.90 | 59.84 | 65min | 165min |
| | 2 | 3 | 5.48 | 5.16 | 3.46 | 44.66 | 45.45 | 60.77 | 100min | 250min |
| ✓ | 2 | 3 | 5.50 | 5.24 | 3.52 | 43.88 | 44.08 | 59.64 | – | – |

**Table 4:** Ablation of refining technology. "w/o confidence means" replacing the confidence score as a consistent value 1.0. "w memory" and "w/o memory" indicate whether using the memory-retrieval mechanism or not. The training time is measured on an NVIDIA V100.

**Results and Analysis** Ablation studies about MSA, recycling and confidence embedding are presented in Table 4. We conclude that:(1) Recycling has a more significant impact on performance than virtual MSA. When increasing the recycling from 1 to 3 while keeping the number of virtual MSAs constant at 1, the recovery rate on the full dataset improves by 1.95%, from 58.39% to 60.34%. In contrast, increasing the number of virtual MSAs from 1 to 3 only results in a 0.32% improvement. (2) The confidence-aware tuning module makes a non-trivial improvement by 0.78%, 1.37%, and 1.13% on the short, single-chain, and full datasets, respectively. (3) Increasing the number of virtual MSAs and recycling times leads to higher computational overhead during training. Therefore, we introduce a memory retrieval mechanism that saves more than 50% of the training time in all cases. (4) Based on the above analysis, we conclude that the importance order of the three components is as follows: recycling > confidence-aware tuning module > virtual MSA.

## 4.5 PRETRAIN KNOWLEDGE(Q3)

**Objective & Setting**    By deleting the corresponding pre-trained embedding in Eq.12, we investigate how much performance gain the model can achieve from different pre-trained models, including ESM2-650M (Meier et al., 2021; Lin et al., 2022), ESM-IF (Hsu et al., 2022), and GearNet (Zhang et al., 2022b). The experimental settings are kept the same as in Section 4.1.

**Table 5:** Ablation study on multimodal knowledge. We investigate how much performance gain the model can obtain from different pre-trained models, where ESM+ESMIF provides the best performance.

| Model | Perplexity ↓ | | | Recovery % ↑ | | |
|---|---|---|---|---|---|---|
| | Short | Single-chain | All | Short | Single-chain | All |
| PiFold | 6.04 | 6.31 | 4.55 | 39.84 | 38.53 | 51.66 |
| KW-Design (GearNet) | 6.66 | 6.89 | 4.96 | 38.72 | 38.02 | 50.43 |
| KW-Design (ESM2-650M) | 6.05 | 5.29 | 3.90 | 43.32 | 46.30 | 57.38 |
| KW-Design (ESMIF) | 6.15 | 6.51 | 4.18 | 38.79 | 39.71 | 54.52 |
| KW-Design (ESMIF+ESM2-650M) | **5.48** | **5.16** | **3.46** | **44.66** | **45.45** | **60.77** |

**Results & Analysis**    We present results on Table.5. We observe that: (1) The knowledge of ESMIF and ESM pre-trained models contributes to improving the performance, with ESM providing a larger improvement than ESMIF. Specifically, the knowledge of ESMIF and ESM results in a 2.86% and 5.72% improvement, respectively. In contrast, the knowledge of GearNet does not contribute to the improvement. (2) The best recovery rate is achieved when the model combines the knowledge of ESM and ESMIF, resulting in a 9.11% improvement. Notably, the improvement is not linear, as the combination of ESMIF and ESM provides a larger improvement than the sum of their individual contributions (i.e., $9.11\% > 2.86\% + 5.72\%$). These results highlight the importance of selecting the appropriate pre-trained models for protein structure refinement and demonstrate the effectiveness of combining multiple sources of knowledge to achieve better performance.

## 4.6 MORE ANALYSIS(Q4)

**Recovery States**    We randomly selected 10 proteins from the CATH4.2 dataset test set and designed their sequences using PiFold. The sequences were then refined using KW-Design, and the recovery states of the designed residues were visualized in Fig. 3. Our results show that KW-Design tends to make more positive corrections than negative corrections, with positive corrections occurring mostly in adjacent locations to initially positive residues. This suggests that the model learns the local consistency of the protein structure and can automatically correct incorrectly designed residues that violate this consistency.

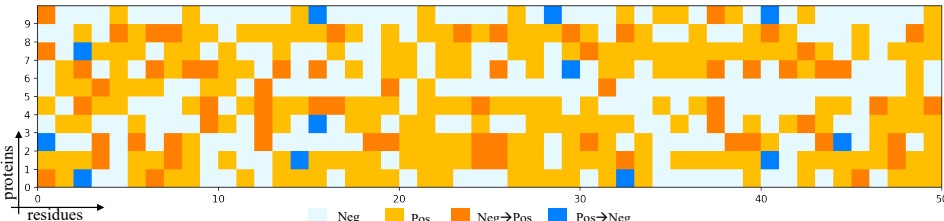

**Figure 3:** Recovery states. The light blue cell indicates a negative residue designed by PiFold and differs from the reference one, while the light orange cell indicates a positive residue that matches the reference one. The darker blue cells indicate where KW-Design wrongly converted positive residues into negative ones, while the darker orange cells indicate where KW-Design corrected negative residues to positive ones.

Due to space limitations, we have included additional analysis and details in the appendix, which will help readers to better understand the proposed method.

## 5 CONCLUSION&LIMITATION

We propose KW-Design, a novel method that iteratively refines low-confidence residues using common protein knowledge extracted from pretrained models. KW-Design is the first model that achieves 60+% recovery on CATH4.2, CATH4.3, TS50, TS500, and PDB, demonstrating its effectiveness and generalizability. However, the proposed method has not yet been verified through wet experiments in real applications, and this will be a direction for future work.

## 6  ACKNOWLEDGEMENTS

This work was supported by National Science and Technology Major Project (No. 2022ZD0115101), National Natural Science Foundation of China Project (No. U21A20427), the Center of Synthetic Biology and Integrated Bioengineering of Westlake University and Integrated Bioengineering of Westlake University Project (No. WU2022A009) and the Westlake University Industries of the Future Research Funding Project (No. WU2023C019).

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

# A MORE ANALYSIS

**Module-wise training** Regarding to the module-wise training strategy, we employ a stagewise fashion for sequence refinement. This is represented by the equation $s^{(l)} = f_{\phi^{(l)}}(s^{(l-1)}; z^{(l-1)})$, where $s^{(l)}$ represents the refined sequence generated by the $l$-th refinement module $f_{\phi^{(l)}}$, and $z^{(l-1)}$ denotes the pretrained embedding extracted from $s^{(l-1)}$. The module-wise training could significantly reduce the computational cost and helpful for alleviating oversmoothing:

1. Stantard: For a given protein, during each epoch, the parameters of layers $l$ and $l + 1$ are updated, resulting in a change in $s^{(l)}$. Due to the large size of the pretrained model (ESM2-650M), performing a forward pass to obtain the updated version of $z^{(l)}$ consumes significant GPU memory and increases the time cost. Consequently, conducting end-to-end training would significantly increase the computational cost.

2. Ours: By fixing $\theta^{(l)}$ before training $\theta^{(l+1)}$, we can ensure that $s^{(l)}$ is retrained without the need for updating $z^{(l)}$. This approach allows us to initialize $s^{(l)}$ in the first epoch and reuse it in subsequent epochs, effectively avoiding redundant forward passes of the large pretrained model.

3. The overall KWDesign is $l * 10$-layer GNN model, where each refinement module consists of 10 GNN layers. It is well known that GNN models suffer from the issue of oversmoothness when training in an end-to-end fashion. To overcome this, we adopt module-wise training to ensure that the pre-placed module serves as a good initialization for the subsequent module. As discovered by (Pina & Vilaplana, 2023), the layer-wise training promotes the node features to be uncorrelated at each single layer to alleviate oversmoothing.

**Distribution comparison** Fig. 4 shows the confidence distributions of positive and negative residues generated by PiFold and KW-Design on the CATH4.2 test set. Positive residues tend towards a confidence of 1.0, while negative residues have mostly below 0.6 confidence, indicated by different colors. Our results demonstrate that KW-Design produces positive residues with higher confidence compared to PiFold, while also reducing the number of negative residues. This suggests that KW-Design can convert low-confidence positive residues to high-confidence ones and correct negative residues as positive ones.

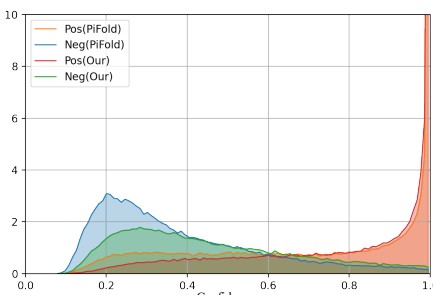

**Figure 4:** Confidence distributions.

**Compare structures** In Fig.5, we use ESMFold(Lin et al., 2022) to generate protein structures from designed sequences and compare the designed proteins of PiFold and KW-Design against the reference ones. We observe that the designed structures of KW-Design are more similar to the reference ones than those of PiFold. Specifically, KW-Design achieves 15.9%, 35.3%, and 60% improvement in structural mean squared error (MSE) on the 1a73, 1a81, and 1ac1 proteins, respectively. These results demonstrate that KW-Design can generate proteins that are structurally more similar to the reference ones compared to PiFold.

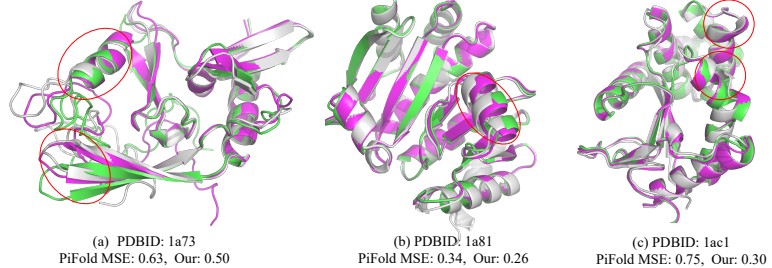

(a) PDBID: 1a73
PiFold MSE: 0.63, Our: 0.50

(b) PDBID: 1a81
PiFold MSE: 0.34, Our: 0.26

(c) PDBID: 1ac1
PiFold MSE: 0.75, Our: 0.30

**Figure 5:** Comparing the designed proteins. The green structures are reference ones, while the gray and purple structures are designed by PiFold and KW-Design, respectively. We use red circles to highlight the regions where KW-Design produces more similar structures to the reference ones than PiFold.

## B    DISCUSSION ABOUT DIVERSITY

**TS45**    In addition to designing single- and multi-chain proteins, we also include a set of *de novo* proteins collected from the CASP15 competition to provide a more realistic assessment (Senior et al., 2019; Kinch et al., 2021). The Critical Assessment of Protein Structure Prediction (CASP15), which took place from May through August 2022, was held after the release dates of CATH4.3 (July 1, 2019) and PDB (August 2, 2021). In CASP15, diverse protein targets are introduced, including FM (Free Modeling), TBM (Template-Based Modeling), TBM-easy, and TBM-hard proteins. There are 18 FM, 25+2 TBM (including 20 TBM-eazy, 5 TMB-hard, 2 FM/TBM). The FM targets have no homology to any known protein structure, making them particularly suitable for *de novo* protein design. The TBM targets have some homology to known protein structures, while the TBM-easy targets are relatively easy TBM targets. The TBM-hard targets are more difficult TBM targets, with lower levels of sequence identity to known structures. We download the public TS-domains structures from CASP15 which consists of 45 structures, namely TS45. We use TS45 as a benchmark for *de novo* protein design, as the structures are less similar to known structures and were not determined prior to the construction of the training sets.

**Diversity Definition**    To improve the success rate of protein design, it is important to explore a set of protein sequences rather than placing a bet on a single sequence. In this case, generating diverse sequences is crucial for exploring the reasonable protein sequence space. We define the pairwise diversity (Jain et al., 2022) as $D_{ij} = \frac{\sum_{l=1}^{n} \mathbb{1}_{r_{i,l} \neq r_{j,l}}}{n}$, where $r_{i,l}$ indicates the $l$-th residue of the $i$-th designed sequence. The overall diversity score is

$$\mathrm{Div} = \sum_{i,j} \frac{D_{i,j}}{m^2} \tag{14}$$

where $i, j \in \{1, 2, 3, \cdots, m\}$ and $m$ is the number of totally designed sequences. By default, we set $m = 10$. However, measuring diversity alone without combining it with other metrics may be misleading. For example, a high diversity indicates a low recovery rate, more likely to result in a low structural similarity.

**Experiments & Analysis**    We benchmark the diversity on TS45 dataset using models pre-trained on CATH4.3. As discovered by previous research (Hsu et al., 2022; Dauparas et al., 2022), the sampling temperature affects diversity. Denote the temperature as $T$, the predicted probability vector is $\boldsymbol{p} \in \mathbb{R}^{n,20}$, we sample new sequences from the distribution of $\mathrm{Multinomial}(\mathrm{softmax}(\boldsymbol{p}/T))$. We vary the temperature from 0.0 to 0.5 and plot the trends of recovery and diversity in Fig. 6. Under the same sampling temperature, high recovery leads to decreased diversity. **However, at the same level of recovery, stronger models have higher diversity.** This phenomenon can be attributed to the fact that stronger models, such as KWDesign, PiFold, and ProteinMPNN, exhibit relatively high confidence levels for a larger number of residues. Even after applying smoothing to the probability distribution, the recovery rate remains consistently high, while a noticeable improvement is observed in terms of diversity.

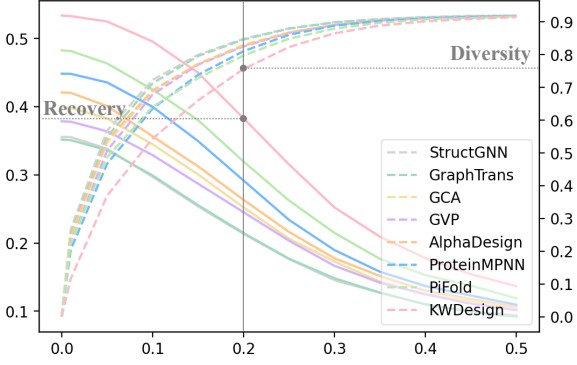

**Figure 6:** The trends of recovery and diversity.

## C  MORE COMPARISON

**Compare to LMDesign**    In parallel with our work, we are observing another exciting project called LMDesign (Zheng et al., 2023), which was recently published at ICML as an oral presentation. LMDesign aims to use the pre-trained ESM model to improve protein design. However, there are several differences between our knowledge-Design and LMDesign.

- **More comprehensive**: We enhance protein design by fusing multimodal knowledge from pre-trained models, including both structural and sequential information, while LMDesign only uses single-modal information. Our experiments demonstrate that combining these modalities leads to nontrivial improvements, as shown in Table 5

- **More efficient**: We introduce the memory-retrieval mechanism to save more than 50% of the training time, while LMDesign does not use this mechanism.

- **Novel modules**: We introduce confidence-aware recycling techniques as well as virtual MSA to boost the model performance.

- **More effective**: Overall, our model outperforms LMDesign by 5.12% on the CATH4.2 dataset.

**Table 6:** Results comparison on the CATH dataset. All baselines are reproduced under the same code framework, except ones marked with †. We copy results of GVP-large and ESM-IF from their manuscripts (Hsu et al., 2022). The **best** and suboptimal results are labeled with bold and underline.

| Model | Perplexity ↓ | | | Recovery % ↑ | | | CATH version | |
|---|---|---|---|---|---|---|---|---|
| | Short | Single-chain | All | Short | Single-chain | All | 4.2 | 4.3 |
| StructGNN | 8.29 | 8.74 | 6.40 | 29.44 | 28.26 | 35.91 | ✓ | |
| GraphTrans | 8.39 | 8.83 | 6.63 | 28.14 | 28.46 | 35.82 | ✓ | |
| GCA | 7.09 | 7.49 | 6.05 | 32.62 | 31.10 | 37.64 | ✓ | |
| GVP | 7.23 | 7.84 | 5.36 | 30.60 | 28.95 | 39.47 | ✓ | |
| GVP-large† | 7.68 | 6.12 | 6.17 | 32.6 | 39.4 | 39.2 | | ✓ |
| AlphaDesign | 7.32 | 7.63 | 6.30 | 34.16 | 32.66 | 41.31 | ✓ | |
| ESM-IF† | 8.18 | 6.33 | 6.44 | 31.3 | 38.5 | 38.3 | | ✓ |
| ProteinMPNN | 6.21 | 6.68 | 4.61 | 36.35 | 34.43 | 45.96 | ✓ | |
| PiFold | 6.04 | 6.31 | 4.55 | 39.84 | 38.53 | 51.66 | ✓ | |
| LMDesign | 6.77 | 6.46 | 4.52 | 37.88 | 42.47 | 55.65 | ✓ | |
| Knowledge-Design (Ours) | **5.48** | **5.16** | **3.46** | **44.66** | **45.45** | **60.77** | ✓ | |

**sc-TM** The structural similarity is the ultimate standard for measuring the quality of the designed sequence. However, the structures of designed protein sequences needed to be predicted by other algorithms, such as AlphaFold (Jumper et al., 2021), OmegaFold (Wu et al., 2022) and ESMFold (Lin et al., 2022). The protein folding algorithm itself has a certain inductive bias and will cause some prediction errors, which will affect the evaluation. To overcome the inductive bias, we introduce the self-consistent TM-score (sc-TM) metric:

$$\text{sc-TM} = \text{TMScore}(f(\hat{\mathcal{S}}), f(\mathcal{S})) \tag{15}$$

where $f$ is the protein folding algorithm and $\text{TMscore}(\cdot, \cdot)$ is a widely used metric (Zhang & Skolnick, 2005) for measuring protein structure similarity. Since the structures of the designed sequence and reference sequence are predicted by the same protein folding algorithm, the model's inductive bias is expected to be canceled out when calculating the TM-score. This approach results in a more robust metric, called the sc-TM, that is less affected by the inductive bias of the protein folding algorithm.

**sc-TM Results** On the CASP15 dataset, we investigate the recovery and sc-TM metrics as the temperature increases. To compute sc-TM, we utilize AlphaFold to predict protein structures from sequences. According to Fig.7 and Fig.6, we observe that a slight increase in temperature from 0 to 0.1 is beneficial in significantly enhancing diversity while maintaining good recovery and sc-TM. Specifically, when the sampling temperature is set to 0.0 and 0.1, KWDesign outperforms the baselines, achieving the highest sc-TMScore.

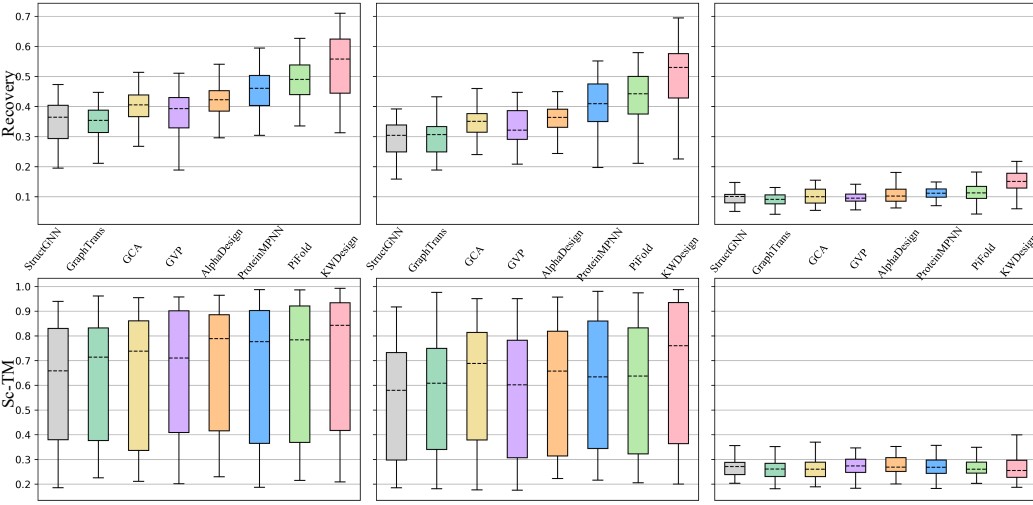

**Figure 7:** The statistics of recovery and sc-TM with increasing temperature.

**Do pretrained language models prefer to correct disordered regions?** Thanks to the insightful comments of Reviewer 4EsM, we investigate the preferred regions that pretrained language models tend to correction. Expeirments follow five steps:

1. Select PDBs containing diverse protein structures, run the pretrained PiFold to get the designed sequence $S_{pi}$

2. Further use the pretrained KWDesign to refine $S_{pi}$, resulting in the optimized sequence $S_{kw}$. All the used PiFold and KWDesign are pretrained on CATH4.3.

3. For each residue, we record whether PiFold has successfully recovered the residue type ($good_{pi}$) or not ($bad_{pi}$). Similarly, we record $good_{kw}$ and $bad_{kw}$.

4. We list four states for each residues and set a different color for each state: $good_{pi} \rightarrow bad_{kw}$(red), $bad_{pi} \rightarrow bad_{kw}$(gray), $good_{pi} \rightarrow good_{kw}$(white), $bad_{pi} \rightarrow good_{kw}$(green).

5. We show the colored proteins with PyMol and present them in Fig.8.

Our finding is that pretraining knowledge corrects for both disordered and ordered regions. However, it seems that the correction is more pronounced for regions with well-defined secondary structures ($\alpha$-helix and $\beta$-sheet).

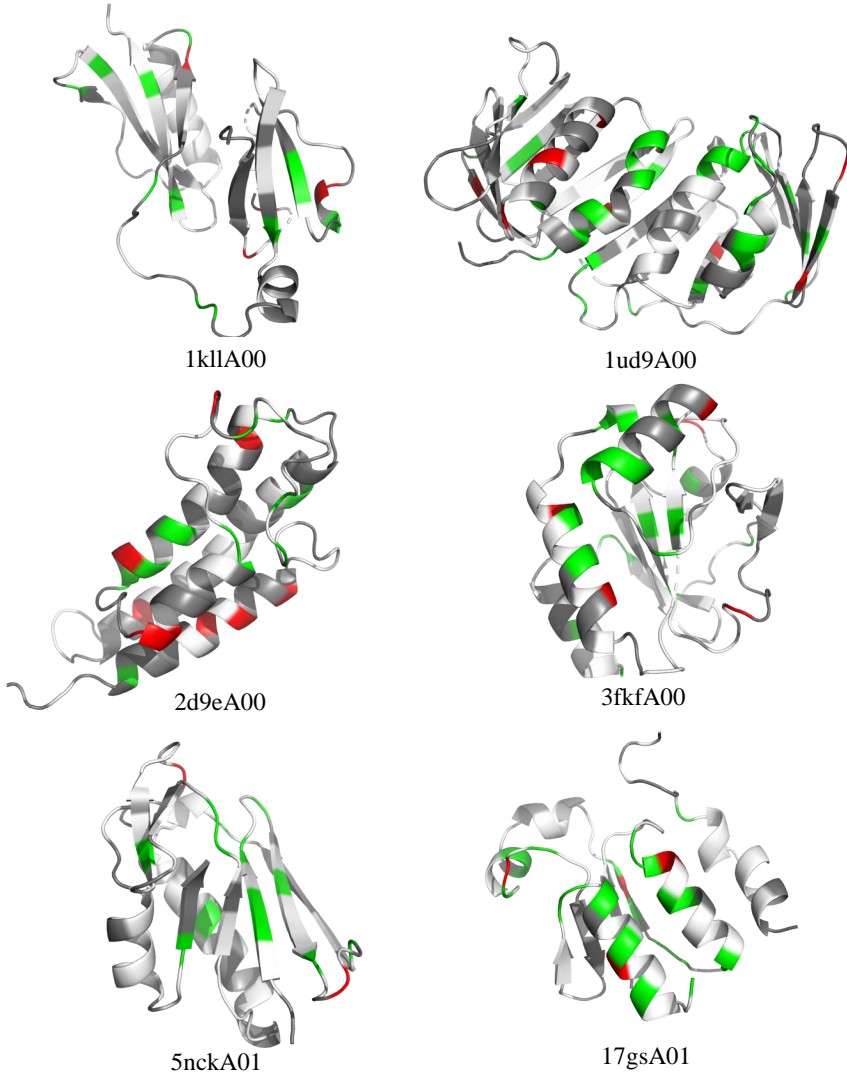

1kllA00      1ud9A00

2d9eA00      3fkfA00

5nckA01      17gsA01

**Figure 8:** The colored proteins.

