# OpenReview forum: "KW-Design: Pushing the Limit of Protein Design via Knowledge Refinement"
_ICLR.cc/2024/Conference — ICLR 2024 poster_

### Official Review · Reviewer_4EsM · 2023-10-26

**Soundness:** 2 fair
**Presentation:** 2 fair
**Contribution:** 3 good
**Rating:** 6
**Confidence:** 5

**Summary:**

The paper addresses the 'inverse protein folding'  problem of designing a protein sequence that folds into a particular shape. They achieve strong results on standard benchmarks.  It draws on a number of interesting ideas, from graph neural networks to leveraging embeddings of pretraining language models, to a 'recycling' approach that updates predictions based on the current uncertainty over those predictions.

**Strengths:**

The paper achieves really strong empirical results on a panel of common benchmarking setups. The results will definitely be of interest to the community.

**Weaknesses:**

The paper's explanation of the model is extremely difficult to understand because it is not using standard terms for deep neural networks. For example, 'knowledge' is used in a vague way, I guess, to mean leveraging pretraining? If I understand correctly, the composition of functions in section 3.1 is just describing a multi-layer neural network, but uses very verbose and non-standard notation. Further, why are the layers trained in a stagewise fashion in section 3.4 instead of standard back-propagation? It was confusing to me why you chose this, since it is much more complex to implement. Does it provide better performance?

The paper's experiments are quite careful about train-test splits, using a number of clustering-based splits that are well-established in the literature. I'm concerned about data leakage, however, from the pretrained language models used to help guide predictions for low-confidence residues in the 'knowledge' module. There models were trained across vast numbers of proteins and likely do not follow the same train-test splits as for the structure -> sequence benchmarks. As a result. The exact target sequence for structure -> sequence design may have appeared in the LM pretraining data. This may explain why the paper's method is able to increase the per-residue recovery rate so dramatically. The paper also mentions that the ESM-IF train-test split is not compatible with some of the other benchmarking setups, yet ESM-IF embeddings are used here.

**Questions:**

Can you please clarify my questions regarding train-test splits and data leakage from the 'knowledge' module?

Can you please explain why you use such a non-standard training procedure?

I have raised my score to weak accept. We had an extensive back and forth regarding train-test splits and I appreciate the attention that the authors have devoted to this important topic.


==Comments after author's response==
My principal concerns about the paper were (1) data leakage and (2) the unconventional presentation of a multi-layer neural network architecture and stagewise training procedure.

Regarding (1), the overall challenge with this research field is that the primary application of inverse folding models is to find sequences that fold into de-novo designed protein structures. However, to validate models on offline natural data, the sequence-structure pairs in the test set aren't de-novo, particularly if they were seen in pretraining. The authors created a new test set at my request, which is small, but establishes good community standards for how to think about  these issues in the future.

Regarding (2), please update the paper to use more standard terminology for a multi-layer model. When describing the stagewise training procedure, please make it clear in updated versions of the manuscript that this was done for computational reasons, and the superiority of it in terms of achieving a better model is not evaluated empirically.

Finally, note that in future work an alternative to your stagewise training approach would be to use gradient checkpointing, which reduces memory overhead (at the expense of additional forward passes).

---

> ### Author Response · Authors · 2023-11-15
> **To reviewer 4EsM**
>
> Dear reviewer:
>
> We greatly appreciate your insightful comments and review service! We are pleased to address your questions and concerns:
>
> > **W0** The paper's explanation of the model is extremely difficult to understand because it is not using standard terms for deep neural networks. For example, 'knowledge' is used in a vague way, I guess, to mean leveraging pretraining?
> >> **Reply to W0**  Regarding the usage of the term **'knowledge'**, it refers to the pretrained embedding utilized in our model. However, it is important to note that different tuning modules do not output the same embedding. This is because the protein sequence undergoes refinement after each tuning layer, requiring updates to these embeddings.
>
> > **W1 & Q1**  If I understand correctly, the composition of functions in section 3.1 is just describing a multi-layer neural network, but uses very verbose and non-standard notation. Further, why are the layers trained in a stagewise fashion in section 3.4 instead of standard back-propagation? It was confusing to me why you chose this, since it is much more complex to implement. Does it provide better performance?
>
> > Can you please explain why you use such a non-standard training procedure?
>
> >> **Reply to W1** Thank you for your valuable questions and feedback!  We appreciate your concerns regarding the non-standard terminology used in our paper and the choice of a stagewise training procedure.
>
> >> Concerning the **non-standard training procedure**, we employ a stagewise fashion for sequence refinement. This is represented by the equation $s^k = f_{\theta^{(k)}}(s^{k-1}; z^{k-1})$, where $s^k$ represents the refined sequence generated by the $k$-th refinement module $f_{\theta^{(k)}}$, and $z^{k-1}$ denotes the pretrained embedding extracted from $s^{k-1}$.
>
> >> **Reduce computation cost** Training the model in a standard back-propagation fashion would impose a significant computational burden [1], as shown in Table 4.
>
> >> 1. Stantard: For a given protein, during each epoch, the parameters of layers $k$ and $k+1$ are updated, resulting in a change in $s^{(k)}$. Due to the large size of the pretrained model (ESM2-650M), performing a forward pass to obtain the updated version of $z^{(k)}$ consumes significant GPU memory and increases the time cost. Consequently, conducting end-to-end training would significantly increase the computational cost.
>
>
> >> 2. Ours: By fixing $\theta^{(k)}$ before training $\theta^{(k+1)}$, we can ensure that $s^{(k)}$ is retrained without the need for updating $z^{(k)}$. This approach allows us to initialize $s^{(k)}$ in the first epoch and reuse it in subsequent epochs, effectively avoiding redundant forward passes of the large pretrained model. In addition, as found in the literature [2], layer-wise training GNN is faster than state-of-the-arts by at least an order of magnitude, with a consistent of memory usage not dependent on dataset size, while maintaining comparable prediction performance.
>
> >> **Alleviate oversmoothing** The overall KWDesign is ($k$*10)-layer GNN model when recycling for $k$ times, where each refinement module consists of 10 GNN layers. It is well known that GNN models suffer from the issue of oversmoothness when training in an end-to-end fashion. To overcome this , we adopt module-wise training to ensure that the pre-placed module serves as a good initialization for the subsequent module. As discovered by [1], they find that the layer-wise training promotes the node features to be uncorrelated at each single layer to alleviate oversmoothing.
>
> >>  Given the above considerations, one of our contributions is the module-wise training strategy for GNNs while using pretrained models.
>
>
> >> [1] Pina, Oscar, and Verónica Vilaplana. "Layer-wise training for self-supervised learning on graphs." arXiv preprint arXiv:2309.01503 (2023).
>
> >> [2] You, Yuning, et al. "L2-gcn: Layer-wise and learned efficient training of graph convolutional networks." Proceedings of the IEEE/CVF conference on computer vision and pattern recognition. 2020.
>
>
> > **Q2**: Can you please clarify my questions regarding train-test splits and data leakage from the 'knowledge' module?
> >> **Reply to Q2** We would like to direct your attention to the common question section where we have provided detailed explanations and clarifications.
>
> We hope these responses adequately address your concerns and provide the necessary clarification.
>
> Best regards,
>
> Authors.

---

> > ### Author Response · Authors · 2023-11-21
> > **Looking for your response**
> >
> > Dear Reviewer 4EsM,
> >
> > We sincerely thank you for taking the time to review our manuscript and providing valuable suggestions.
> >
> > Unlike previous years, there will be no second stage of author-reviewer discussions this year, and a recommendation needs to be provided by November 22, 2023. We hope we have satisfactorily addressed your concerns. If so, could you please consider increasing your rating? If you still need any clarification or have any other concerns, please feel free to contact us and we are happy to continue communicating with you.
> >
> > Best,
> >
> > Authors

---

### Official Review · Reviewer_KJsj · 2023-10-29

**Soundness:** 3 good
**Presentation:** 3 good
**Contribution:** 3 good
**Rating:** 6
**Confidence:** 3

**Summary:**

The paper introduces KW-Design, a novel method for protein design that iteratively refines low-confidence residues using knowledge extracted from pretrained models. The approach incorporates a multimodal fusion module, virtual MSA, recycling technologies, and a memory-retrieval mechanism to enhance performance and efficiency. The method demonstrates substantial improvements across various benchmarks, achieving over 60% recovery on datasets such as CATH4.2, CATH4.3, TS50, TS500, and PDB.

**Strengths:**

1. The paper is well written and exhibits a clear and logical structure.
2. The proposed method effectively leverages the knowledge from pretrained protein sequence/structure models, resulting in notable benefits for protein sequence design.
3. The paper includes a thorough ablation study, examining different components of the models such as recycling, virtual MSA numbers, and the pretrained model. This is crucial for gaining a deep understanding of the proposed methodology.

**Weaknesses:**

1. The code associated with the paper is currently unavailable.
2. Given that the model relies on pretrained models, some of which have been trained on the test set utilized, there is a potential risk of data leakage.
3. The paper predominantly employs perplexity and recovery as metrics for evaluating the designed sequences. However, there is a chance that the designed proteins may not be soluble or may not fold correctly to the given backbone. It would be beneficial for the authors to incorporate additional metrics (e.g., scTM score, solubility) in their evaluation.

**Questions:**

1. Is there any fine-tuning done on the pretrained language model used in your approach?
2. In Section 4.3, Table 3 claims that “the predictive confidence score, an unsupervised metric, exhibits a strong correlation with recovery.” Could the authors provide a more detailed analysis, perhaps including the Spearman correlation between these two values?
3. Regarding the virtual MSA, does the sequence order affect the resulting residue embedding? If not, what criteria are used to determine the sequence order?
4. In Section 3.3, the initial node and edge features are extracted using PiFold. Is PiFold fine-tuned or kept fixed during this process?

---

> ### Author Response · Authors · 2023-11-15
> **To reviewer KJsj**
>
> Dear reviewer:
>
> Thanks for your insightful comments! We appreciate the opportunity to answer your questions:
>
> > **W1**: The code associated with the paper is currently unavailable.
> >> **Reply to W1** During rebuttal, we deploy KWDesign to Colab for your check. It is on the way.
>
> > **W3**: The paper predominantly employs perplexity and recovery as metrics for evaluating the designed sequences. However, there is a chance that the designed proteins may not be soluble or may not fold correctly to the given backbone. It would be beneficial for the authors to incorporate additional metrics (e.g., scTM score, solubility) in their evaluation.
> >> **Reply to W3** Thanks for your good question. In the updated appendix, we compared the scTM score of different methods, where KWDesign consistently outperformed baselines.
>
> > **Q1** Is there any fine-tuning done on the pretrained language model used in your approach?
> >> **Reply to Q1** The pretrained language models are frozen.
>
> > **Q2** In Section 4.3, Table 3 claims that “the predictive confidence score, an unsupervised metric, exhibits a strong correlation with recovery.” Could the authors provide a more detailed analysis, perhaps including the Spearman correlation between these two values?
> >> **Reply to Q2** Thanks for your insightful comments. To clarify, higher confidence generally correlates with higher recovery rates. Taking PiFold and KWDesign as examples, we analyze recovery statistics within specific confidence ranges. In the test set of CATH4.2, we record the confidence and recovery (0 or 1) for each residue in all proteins. Finally, we calculate the average recovery for residues falling within a particular confidence range, shown as follows:
>
> >>| conf        | >0.0   | >0.1   | >0.2   | >0.3   | >0.4   | >0.5   | >0.6   | >0.7   | >0.8   | >0.9   | >0.95  |
> >>|----------|-------|-------|-------|-------|-------|-------|-------|-------|-------|-------|-------|
> >>| PiFold   | 51.58 | 51.60 | 55.43 | 63.04 | 69.87 | 75.99 | 81.75 | 86.66 | 90.96 | 95.09 | 97.44 |
> >>| KWDesign | 60.25 | 60.26 | 62.08 | 67.16 | 72.71 | 78.00 | 83.08 | 87.63 | 91.76 | 95.49 | 97.48 |
>
> >> The table shows that higher confidence generally correlates with higher recovery rates.
>
> > **Q3** Regarding the virtual MSA, does the sequence order affect the resulting residue embedding? If not, what criteria are used to determine the sequence order?
> >> **Reply to Q3** As shown in Eq.(12), we employ a summation operation to merge features of virtual MSA sequences. Consequently, the order of the sequences does not affect the residue embedding. In other words, the residue embedding remains unaffected by the sequence order.
>
> > **Q4** In Section 3.3, the initial node and edge features are extracted using PiFold. Is PiFold fine-tuned or kept fixed during this process?
> >> **Reply to Q4** PiFold kept fixed during the process.
>
> We hope these responses adequately address your concerns and provide the necessary clarification.
>
> Best regards,
>
> Authors.

---

> > ### Author Response · Authors · 2023-11-21
> > **Looking for your response**
> >
> > Dear Reviewer KJsj,
> >
> > We sincerely thank you for taking the time to review our manuscript and providing valuable suggestions.
> >
> > Unlike previous years, there will be no second stage of author-reviewer discussions this year, and a recommendation needs to be provided by November 22, 2023. We hope we have satisfactorily addressed your concerns. If so, could you please consider increasing your rating? If you still need any clarification or have any other concerns, please feel free to contact us and we are happy to continue communicating with you.
> >
> > Best,
> >
> > Authors

---

> > > ### Comment · Reviewer_KJsj · 2023-11-22
> > > **Thanks for your reply**
> > >
> > > Thank you for your comprehensive rebuttal. You have addressed most of my concerns. Considering the novelty, experimental rigor, and methodological soundness of your work, I am convinced of its quality and will maintain my rating of 6.

---

> ### Author Response · Authors · 2023-11-22
> **Thank you!**
>
> We are delighted to learn that you have found most of your concerns addressed and that you are maintaining your rating of 6 at this stage. Your recognition of the novelty, experimental rigor, and methodological soundness of our research is greatly appreciated.
>
> We would respectfully inquire if there might be any additional opportunities for us to further improve our manuscript and potentially increase the rating. We remain fully committed to continuously enhancing the quality of our work.
>
> Once again, we express our sincere appreciation for your valuable feedback. If you have the time, we would eagerly welcome any further guidance you may provide. Thanks for your patience, review efforts, and good questions!

---

### Official Review · Reviewer_Mme4 · 2023-11-29

**Soundness:** 3 good
**Presentation:** 1 poor
**Contribution:** 2 fair
**Rating:** 6
**Confidence:** 3

**Summary:**

This work proposes a refining method, called KW-design, to refine the generated protein sequences from previous methods, such as PiFold. In specific, KW-design is an iterative process that uses pretrained sequence and structure models to extract multi-modal features and a confidence-aware gated layer to fuse these features. It also applies a memory retrieval mechanism to save the training time.

**Strengths:**

- The idea of iteratively refining generated sequences from protein structures based on a confidence-aware gated fusion layer that fuses features from multi-modal feature extractors looks novel to me.
- Experiments across multiple datasets show the effectiveness of the proposed method.
- Ablation studies are well conducted to show the impact of main components.

**Weaknesses:**

My main concern is on the clarification of the paper. I feel like a large revision is needed to make the presentation more clear to the readers.
- The method section is not very clear. In particular, Section 3.2 and 3.3 have a large portion of redundancy. For instance, I think “Knowledge extractor & Confidence predictor” in Section 3.3 seems to just repeat the content in “Virtual MSA” in Section 3.2. “Fuse layer” can be merged to “Virtual MSA” to make the presentation more concise. Besides, I don’t know how Eq (13) is related to Eqs (8) and (9). It looks like they are both about how to update the embeddings, but how does “PiGNN” process $e^{(l)}$ to make Eq. (13) consistent to Eqs (8) and (9)?
- In “Confidence-aware updating”, why do you need two MLPs to process the same input (one is the negative sign of another) for the gates? Any justification on Eq. (11)?
- In Algorithm 1, I’m not sure how the optimal $\phi^{(l)}$ is determined and how the “patience of 5” is applied. Also, in line 12 (Algorithm 1), what does $n$ mean in $h_n^{(l+1)}$?
- I wonder why not split Table 1 into two subtables: one for CATH 4.2 and another for CATH 4.3?
- In Table 2, what does the “Worst” metric mean? Why does the proposed method perform worse than all other methods?
- Minor typo issue. “settings Specifically” -> “settings. Specifically” on page 6. “3d CNN” -> “3D CNN” on page 3.

**Questions:**

See my comments in the above.

---

> ### Author Response · Authors · 2023-11-30
> **Author rebuttal**
>
> Dear reviewer Mme4,
>
> We thanks for your careful review and efforts in helping us improving the amnuscript!
>
> > **Q1.1** The method section is not very clear. In particular, Section 3.2 and 3.3 have a large portion of redundancy. For instance, I think “Knowledge extractor & Confidence predictor” in Section 3.3 seems to just repeat the content in “Virtual MSA” in Section 3.2. “Fuse layer” can be merged to “Virtual MSA” to make the presentation more concise.
>
> >> **Reply to Q1.1** Thank you for your valuable feedback. We are not intended to confuse readers. Instead, we have structured the method section hierarchically: Section 3.2 providesa high-level overview of the computing framewor; Section 3.3 delves into  the specifics of the knowledge-tuning module. The abstract-to-detailed approach result in some redundancy. Following your suggestions, we have simplified the sentences of Section 3.3. The introduction of the knowledge extractor and confidence predictor in Section 3.3 serves to ensure complete description of the knowledge-tuning module.
>
> > **Q1.2** Besides, I don’t know how Eq (13) is related to Eqs (8) and (9). It looks like they are both about how to update the embeddings, but how does “PiGNN” process to make Eq. (13) consistent to Eqs (8) and (9)?
> >> **Reply to Q1.2** We appreciate your suggestion! We have revised the manuscript accordingly. In the revised version, we clarify that PiGNNs function as the refine module in Eq. (9), which is responsible for transforming $z^{(l)}$ into $h^{(l+1)}$.
>
> > **Q2** In “Confidence-aware updating”, why do you need two MLPs to process the same input (one is the negative sign of another) for the gates? Any justification on Eq. (11)?
> >> **Reply to Q2** The two MLPs in our method serve the purpose of computing positive and negative scores to control the updating or retaining of residue embeddings.
>
> >> - As the openreview could not render complex latex code, we rewrite $b = c^{(l+1)}_{s'}$,  $a = c^{(l)}_s$, representing the confidence vectors of refined sequences at the $l+1$ and $l$ knowledge-tuning layer, respectively.
> >> - In Fig. 3, we demonstrate that not all refinements have an positive impact on improving recovery. To address this, we have designed a gated filter that selectively activates the latest refinement or retrain the previous residue embeddings.
> >> - In Eq.(11), $score^+ = \sigma(MLP_1 (b-a))$ measure the superiority of the latest refined embedding over the previous embedding. Similarly, we compute the negative score $score^- = \sigma(\text{MLP}_2 (a-b))$. We use different MLPs to calculate positive and negative scores because they represent different meanings in filtering refined embeddings $h^{(l+1)}$ or   previous embeddings $h^{(l)}$, respectively.
> >> - Finally, we combine the refined residue embeddings and previous embeddings via Eq.(11), that is $ h^{(l+1)} \leftarrow h^{(l+1)} \odot score^+ + h^{(l)} \odot score^-$
> >> We hope this clarification addresses your queries.
>
> > **Q3** In Algorithm 1, I’m not sure how the optimal $\phi^{(l)}$ is determined and how the “patience of 5” is applied. Also, in line 12 (Algorithm 1), what does $n$ mean in $h_n^{(l+1)}$ ?
>
> >> **Reply to Q3**  Thank you for your question. In Algorithm 1, the determination of the optimal $\phi^{(l)}$ is done using a patience of 5. Here's how it works:
> >1. We train the knowledge-tuning modules sequentially, updating $\phi^{(l)}$ at each training process $l$.
> >2. During the training process of the $l$-th knowledge-tuning module, we monitor the average predictive confidence over the training set as a metric for evaluating the performance.
> >3. "Early stopping with a patience of 5" means that if the average predictive confidence does not improve for 5 consecutive training epochs, we stop the training process. The optimal $\phi^{(l)}$ is dertermined by the maximum confidence score.
>
> > In Algorithm 1, $h_n^{(l+1)}$ shoud be $h_b^{(l+1)}$, we have revised the manuscipt accordingly. Thank you for bringing it to our attention.
>
> > **Q4** I wonder why not split Table 1 into two subtables: one for CATH 4.2 and another for CATH 4.3?
> >> **Reply to Q4** We put the results in the same tabel for saving the page space. As you can see, many results and analysis have been removed to the appendix due to the limted space.
>
> > **Q5** In Table 2, what does the “Worst” metric mean? Why does the proposed method perform worse than all other methods?
> >> **Reply to Q5** In Table 2, the "Worst" metric refers to the lowest recovery rate achieved on the test set, following the setting of PiFold. In the case of TS500, all the method could not achieve resonable recovery. It is possible that there are low-quality or difficult structures in the dataset.
>
> > **Q6** Minor typo issue. “settings Specifically” -> “settings. Specifically” on page 6. “3d CNN” -> “3D CNN” on page 3.
> >> **Reply to Q6** Thank you for pointing out the minor typos. We have made the necessary revisions in the manuscript.
>
> Best regards,
>
> Authors.

---

> ### Author Response · Authors · 2023-11-30
> **Author rebuttal**
>
> Dear reviewer Mme4,
>
> We thanks for your careful review and efforts in helping us improving the amnuscript!
>
> > **Q1.1** The method section is not very clear. In particular, Section 3.2 and 3.3 have a large portion of redundancy. For instance, I think “Knowledge extractor & Confidence predictor” in Section 3.3 seems to just repeat the content in “Virtual MSA” in Section 3.2. “Fuse layer” can be merged to “Virtual MSA” to make the presentation more concise.
>
> >> **Reply to Q1.1** Thank you for your valuable feedback. We are not intended to confuse readers. Instead, we have structured the method section of KW-Design hierarchically, with Section 3.2 providing a high-level overview of the computing framework and Section 3.3 delving into the specifics of the knowledge-tuning module. The abstract-to-detailed approach result in some redundancy. Following your suggestions, we have simplified the sentences of Section 3.3. The introduction of the knowledge extractor and confidence predictor in Section 3.3 serves to ensure complete description of the knowledge-tuning module.
>
> > **Q1.2** Besides, I don’t know how Eq (13) is related to Eqs (8) and (9). It looks like they are both about how to update the embeddings, but how does “PiGNN” process to make Eq. (13) consistent to Eqs (8) and (9)?
> >> **Reply to Q1.2** We appreciate your suggestion! We have revised the manuscript accordingly. In the revised version, we clarify that PiGNNs function as the refine module in Eq. (9), which is responsible for transforming $z^{(l)}$ into $h^{(l+1)}$.
>
> > **Q2** In “Confidence-aware updating”, why do you need two MLPs to process the same input (one is the negative sign of another) for the gates? Any justification on Eq. (11)?
> >> **Reply to Q2** The two MLPs in our method serve the purpose of computing positive and negative scores to control the updating or retaining of residue embeddings.
>
> >> - As the openreview could not render complex latex code, we rewrite $b = c^{(l+1)}_{s'}$,  $a = c^{(l)}_s$, representing the confidence vectors of refined sequences at the $l+1$ and $l$ knowledge-tuning layer, respectively.
> >> - In Fig. 3, we demonstrate that not all refinements have an positive impact on improving recovery. To address this, we have designed a gated filter that selectively activates the latest refinement or retrain the previous residue embeddings.
> >> - In Eq.(11), $score^+ = \sigma(MLP_1 (b-a))$ measure the superiority of the latest refined embedding over the previous embedding. Similarly, we compute the negative score $score^- = \sigma(\text{MLP}_2 (a-b))$. We use different MLPs to calculate positive and negative scores because they represent different meanings in filtering refined embeddings $h^{(l+1)}$ or   previous embeddings $h^{(l)}$, respectively.
> >> - Finally, we combine the refined residue embeddings and previous embeddings via Eq.(11), that is $ h^{(l+1)} \leftarrow h^{(l+1)} \odot score^+ + h^{(l)} \odot score^-$
> >> We hope this clarification addresses your queries.
>
> > **Q3** In Algorithm 1, I’m not sure how the optimal $\phi^{(l)}$ is determined and how the “patience of 5” is applied. Also, in line 12 (Algorithm 1), what does $n$ mean in $h_n^{(l+1)}$ ?
>
> >> **Reply to Q3**  Thank you for your question. In Algorithm 1, the determination of the optimal $\phi^{(l)}$ is done using a patience of 5. Here's how it works:
> >1. We train the knowledge-tuning modules sequentially, updating $\phi^{(l)}$ at each training process $l$.
> >2. During the training process of the $l$-th knowledge-tuning module, we monitor the average predictive confidence over the training set as a metric for evaluating the performance.
> >3. We use early stopping with a patience of 5, which means that if the average predictive confidence does not improve for 5 consecutive training iterations, we stop the training process for that specific $l$-th module.
>
> > In Algorithm 1, $h_n^{(l+1)}$ shoud be $h_b^{(l+1)}$, we have revised the manuscipt accordingly. Thank you for bringing it to our attention.
>
> > **Q4** I wonder why not split Table 1 into two subtables: one for CATH 4.2 and another for CATH 4.3?
> >> **Reply to Q4** We put the results in the same tabel for saving the page space. As you can see, many results and analysis have been removed to the appendix due to the limted space.
>
> > **Q5** In Table 2, what does the “Worst” metric mean? Why does the proposed method perform worse than all other methods?
> >> **Reply to Q5** In Table 2, the "Worst" metric refers to the lowest recovery rate achieved on the test set, following the setting of PiFold. In the case of TS500, all the method could not achieve resonable recovery. It is possible that there are low-quality or difficult structures in the dataset.
>
> > **Q6** Minor typo issue. “settings Specifically” -> “settings. Specifically” on page 6. “3d CNN” -> “3D CNN” on page 3.
> >> **Reply to Q6** Thank you for pointing out the minor typos. We have made the necessary revisions in the manuscript.
>
> Best regards,
>
> Authors.

---

### Author Response · Authors · 2023-11-15
**To all reviewers and common concerns**

We thank the reviewers for their insightful and constructive reviews of our manuscript. We are encouraged to hear that the reviewers found our work is well written and exhibits a clear and logical structure. (Reviewers KJsj) and that they think our methodology is effective (Reviewers KJsj, 4EsM). Also, they think that our experiments are comprehensive (Reviewers KJsj) and interesting (Reviewers 4EsM).  In rsponse to feedback, we provide responses below to address each reviewer’s concerns point by point. The response mainly includes:

**New Results on CASP15** We provide new results on the dataset of CASP15 [1] to address the concern of data leakage.

**Source code** During rebuttal, we deploy KWDesign to Colab for your check. It is on the way. Refer to the colab demo at this anonymous link: https://colab.research.google.com/drive/1b8sUgO21uxO8NhQbIlvUZZq339aIhC1C?usp=sharing

**Point to Point Response** We address reviewer's concerns point by point.

Thank you again for all the efforts that help us improve the manuscript. In case our answers have justifiably addressed your concerns, we respectfully hope that you can increase your score to support the acceptance of our work.

## Common concern
> **Data leakage**: reviewer KJsj: Given that the model relies on pretrained models, some of which have been trained on the test set utilized, there is a potential risk of data leakage.

> reviewer 4EsM: I'm concerned about data leakage, however, from the pretrained language models used to help guide predictions for low-confidence residues in the 'knowledge' module. There models were trained across vast numbers of proteins and likely do not follow the same train-test splits as for the structure -> sequence benchmarks. As a result. The exact target sequence for structure -> sequence design may have appeared in the LM pretraining data. This may explain why the paper's method is able to increase the per-residue recovery rate so dramatically. The paper also mentions that the ESM-IF train-test split is not compatible with some of the other benchmarking setups, yet ESM-IF embeddings are used here.


>> **Reply** We really appreciate the good question. We answer the question from several aspects:

>> 1. It is widely accepted in protein sequence design to utilize large pretrained language models, which undergo unsupervised pretraining. For instance, LMDesign[2], presented at ICML2023 as an oral presentation, leverages pretrained language models to enhance structure-to-sequence performance. KW-Design follows the similar setting as LMDesign.

>> 2. In regard to the ESM-IF, we conducted experiments on CATH4.3 using the same data split as its original paper. Table 1 demonstrates that our method outperforms previous state-of-the-art approaches by a significant margin.

>> 3. In traditional benchmarks such as CATH4.2, CATH4.3, TS50, TS500, and PDB, it is difficult to elucidate the impact of the so-called "data leakage" issue when using ESM model. To provide further clarification, we evaluated the trained models on proteins from CASP15. Note that CASP15 is holded after the release date of CATH4.3 and PDB. From May through August 2022, CASP organizers have been posting on this website sequences of unknown protein structures. CATH4.3 uses PDB proteins as of July 1, 2019. ProteinMPNN's PDB dataset is as of 2 August 2021. We believe that this comparison would avoid the issue of data leakage. As shown in the following table, KWDesign consistently outperforms previous methods by a large margin.


| Training Set |Test Set| StructGNN | GraphTrans | GCA  | GVP  | AlphaDesign | ProteinMPNN | PiFold | KWDesign |
|--------------|-----------|-----------|------------|------|------|-------------|-------------|--------|----------|
| CATH4.2      | CASP15 |  0.35      | 0.36       | 0.40 | 0.39 | 0.42        | 0.44        | 0.47   | 0.54     |
| CATH4.3      | CASP15 | 0.36      | 0.35       | 0.41 | 0.39 | 0.42        | 0.46        | 0.49   | 0.56     |
| PDB          | CASP15 | 0.41      | 0.41       | 0.43 | 0.43 | 0.46        | 0.52        | 0.53   | 0.59     |


[1] https://predictioncenter.org/casp15/index.cgi

[2] Zheng, Zaixiang, et al. "Structure-informed Language Models Are Protein Designers." (2023).


## Finally, we highlight our unique contributions:
- **Novelty**: We are the first to cleverly fuse multimodal pretrained models (sequence+structure) to enhance the protein inverse folding.
- **Performance**: KWDesign is the first method that achieves 60+% recovery on single-chain protein design (CATH, TS50, TS500), multi-chain protein design (PDB), and de-nove protein design (CASP15).
- **Data leakage**: Regarding to using pretrained models, we are the first to confront the issue of data leakage and provide results on the CASP15 dataset to address this concern.
- **Comprehensive metrics**: In addition to the recovery, we also analyzed the diversity and scTM in the appendix.

---

> ### Comment · Reviewer_4EsM · 2023-11-21
> **I am still concerned about data leakage**
>
> Thanks for your detailed response. Here is my summary of the my thoughts on the leakage topic. Please respond.
>
> 1) Two kinds of data are used for training models: paired data S = {(sequence, structure)}  and 'pretraining' sequences P = {sequence}.
> 2) The community has established rigorous methods for splitting S into train-test splits that test for strong extrapolation across folds, etc.
> 3) Your paper follows the standards from (2), but also includes P during training.
> 4) There are sequences that appear in the test set of S that also appear in P. Even using CASP15 for S does not prevent this.
> 5) One method for evaluating your models' predictions is 'sequence recovery', the fraction of amino acids that were correctly predicted.
> 6) For structure prediction, generally there are some residues that are easy to predict from the structure and some where there is fundamental uncertainty given the structure (e.g., in disordered regions or in regions with low conservation).
> 7) Roughly speaking, the logic of (6) allows for a given (structure, sequence) pair in the test set to partition the residues in the sequence into subsets R_predictable and R_unpredictable.
> 8) If the sequence in (7) appears in the pretraining data P, the sequence-level pretrained model provides significant information about P(R_unpredictable | R_predictable) (because high-capacity models generally memorize the training data very well).
> 9) The signal from (8) can be used to significantly improve the sequence recovery metric.

---

> ### Author Response · Authors · 2023-11-22
> **Further response to reviewer 4EsM, thanks for your thoughts!**
>
> Dear reviewer 4EsM, we really thanks for your insightful comments! Your thoughts are clear, in-depth and constructive, which are helpful for the field in understanding why pretrained models could enhance protein inverse folding. Response to your efforts, we have conducted additional experiments to clarify them. We will appreciate it if reviewers check the updated appendix that contains rebuttal-related tables and figures.
>
>
> - Firstly, we response to your comprehensive thoughts. We agree with (1,2,3,5,6,9), while suggesting to improve (4,7,8).
>
> - Regarding to (7,8),  basically, we agree with your key insight that pretrained language models are helpful in determing so-called "unpredictable" residues based on "predictable" ones. However, we suggest revising $P(R_{unpredictable} | R_{predictable})$ as $P(R_{hard} | R_{easy})$, as not all successfully refined residues belong to disordered regions or regions with low conservation. We provide visual analysis in the updated **appendix (Figure 8)**. Our finding is that pretraining knowledge corrects for both disordered and ordered regions. However, it seems that **the correction is more pronounced for regions with well-defined secondary structures ($\alpha$-helix and $\beta$-sheet)**. Consequently, one of our contributions is **explicitly considering the difficulty represented by the predictive scores (Pseudo Labels)** in the design of the refinement module.
>
>
>
> - Regarding to (4), we think CASP15 is the best practice currently available for evaluating  inversefolding methods that incorporate knowledge modules, where they may use pretrained models or retrievaled information from external database. From a standard machine learning perspective, it is difficult to find a rigorous test set, as the UniProt database used for training protein language models is extremely large. Fortunately, the CASP15 is holded after than the UniProt version used for ESM pretraining, and the released proteins are regarded by biologists as unknown stuctrues for attracting scientists around the world to participate in structure prediction challenges.  From a practical standpoint, we can employ the **"time split" strategy** to construct the test set. This involves selecting a specific timepoint, where data before that point are assigned to the training set, while data after that point constitute the test set. Notably, such a "time split" strategy has been recognized by some notable works in protein modeling fields, e.g., AlphaFold[1] and RoseTTAFold[2] are trained on previously known proteins and evaluated on CASP14.
>
> [1] Jumper, John, et al. "Highly accurate protein structure prediction with AlphaFold." Nature 596.7873 (2021): 583-589.
>
> [2] Baek, Minkyung, et al. "Accurate prediction of protein structures and interactions using a three-track neural network." Science 373.6557 (2021): 871-876.
>
>
> - Respectfully, we would like to pose the question: should we accept the new setting that utilizes pretrained models, even though LMDesign has already been accepted as an oral representation at ICML2023? The use of foundational models as components is widely accepted in other machine learning domains, such as computer vision and natural language processing. It is worth noting that these fields may also encounter similar concerns regarding data leakage under a strict perspective.  Given that using pretrained models have become a research trend, we cannot prevent their usage entirely, but we can strive to keep the evaluations reasonable. With this in mind, we present our results on the CASP15 dataset.
>
>
> Once again, we sincerely appreciate the reviewers' insightful comments and remain committed to continuously provide additional clarifications. We understand your concern and hope that our response could  shed light on this issue to promote healthy development of protein inverse folding. In the era of large-scale modeling, we would like to say that it seems to contradict current research trends if we reject the new setting created by LMDesign. If we could accept the new setting, we need new evaluation protocol to address the data leakage issue. **Our proposal is to use the "time-split" strategy.** We also look for your suggestions and will take them to better clarify this issue. We are pleased to discuss with you. Thanks for your patience and review efforts!

---

> > ### Comment · Reviewer_4EsM · 2023-11-22
> >
> > This is an important topic, and I'm glad we're discussing it in detail.
> >
> > The main reason that this paper would be accepted is that the performance improvement over baseline methods is quite large. However, it is important to confirm that this is a real performance improvement and not a consequence of the evaluation setup that provides an unfair advantage to your technique.
> >
> > I agree that modern setups that involve pretraining can make train-test splits difficult. I also agree that things get complicated/frustrating when previously-accepted papers used a particular evaluation setup, and then you receive feedback questioning the validity of the setup.
> >
> > Here is my current understanding of the question of data leakage:
> > 1) We agree that modern high-capacity pretrained models can memorize training examples, so they can reconstruct sequences with high residue-level accuracy when provided with a subset of the residues. This makes evaluation difficult if the test-set sequences appeared in pretraining.
> > 1) We agree that there is a significant data leakage opportunity for all setups other than CASP15, since exact sequences in the evaluation set appeared in the pretraining data.
> >
> > In your response, the CASP15 setup is described as a gold standard that gets around the data leakage issue. However, I'd like additional clarification regarding some details. First of all, it is a temporal split in terms of whether crystallography experiments have been done, but not necessarily a temporal split in terms of whether the sequences appear in uniprot. For the CASP-15 test set, can you please inspect some of these entries? Did the sequences first appear in uniprot recently, or have they been in uniprot for a long time (importantly, since before ESM was trained)?
> >
> > Regarding LMDesign, the review process can be noisy, and we should not rely on that paper being accepted as evidence that the setup in your paper is valid for evaluation.

---

> ### Author Response · Authors · 2023-11-26
> **Further clarification**
>
> Thank you for your insightful comments! We appreciate the opportunity to discuss this issue in-depth to enhance the reader's understanding of our work.
>
> ## Concern:
> Reviewer 4EsM: It is important to confirm that this is a real performance improvement and not a consequence of the evaluation setup that provides an unfair advantage to your technique.
>
> ## Agreements:
> 1. We cannot prevent the research trend created by LMDesign that uses pretrained model for protein design tasks.
> 2. We agree that there is a significant data leakage opportunity for all setups other than CASP15, since exact sequences in the evaluation set appeared in the pretraining data.
> 2. Modern setups that involve pretraining can make train-test splits difficult, as we don't know the exact training data of ESM2 model. ESM2's authors say that UniRef50, September 2021 version, is used for the training of ESM models [1] and the dataset was randomly partitioned. Unfortunately, they do not make the data splitting files publically available.
> 3. We should carefully re-examine and and standardize the evaluation protocol.
>
> [1] Lin, Zeming, et al. "Evolutionary-scale prediction of atomic-level protein structure with a language model." Science 379.6637 (2023): 1123-1130.
>
> ## Proposal:
> 1. Our proposal is to implement the **"time-split" strategy**, where data before a specific timepoint are assigned to the training set, and data after that point form the test set. This strategy has been recognized and utilized in notable works such as AlphaFold and RoseTTAFold for evaluating protein modeling methods. We believe it is the current best practice for evaluating inverse folding methods incorporating knowledge modules.
>
> 2. We appreciate the constructive suggestion from reviewers to address this issue.
>
> ## Experiments:
> 1. Structure "time-split": We evaluate pretrained models on the CASP15 dataset to address the concern about structure pretraining model (ESM-IF). The crystal structures in this dataset are novel and not have not been seen before pretraining ESM-IF. KWDesign consistently outperform baselines. However, we cannot guarantee that the sequences were unknown prior to pretraining ESM2-650M, which requires further clarification.
>
> 2. Sequence-Structure "time-split": For further clarification, we plan to create a dataset called NovelPro, which strictly follows a "time-split" approach for both structure and sequence. We have mutually collected 80 recently released protein sequences from SeqRef[2], released within the last 30 days before November 23, 2023. Using AlphaFold2, we predict the structures for these sequences (74 successful, 6 failed). Based on these structures, we employ models pretrained on CATH4.3 to design protein sequences. The structure prediction process takes approximately 2-3 days. We filter a subset of protein structures based on the average pLDDT and report the median recovery:
>
> | Filter | Test Set |   StructGNN | GraphTrans | GCA  | GVP  | AlphaDesign | ProteinMPNN | PiFold |LMDesign| KWDesign |
> |--------------|-----------|-----------|------------|------|------|-------------|-------------|--------|----------|----------|
> | pLDDT>70     |  NovelPro  | 0.40   | 0.40      | 0.43 | 0.42 | 0.46        | 0.52        | 0.57  | 0.59 | **0.64**     |
> | pLDDT>80  |  NovelPro   | 0.40      | 0.41       | 0.45 | 0.43 | 0.48        | 0.54        | 0.58 |  0.60  | **0.66**     |
> | pLDDT>90  |   NovelPro    | 0.43      | 0.43       | 0.47 | 0.45 | 0.49        | 0.57        | 0.60  |  0.62 | **0.68**     |
>
> We use the official code and published checkpoints of LMDesign+ESM2-650M [3], and the reproduced recovery is 55.6 on CATH4.2 and 59.71/59.81 on PDB, which are consistent with the results in their paper. Then, we report the best LMDesign' results on NovelPro, where temperatures and recycling numbers are carefully tuned. The created NovelPro dataset will be released upon acceptance.
>
> [2] https://www.ncbi.nlm.nih.gov/protein?term=srcdb_refseq[prop]%20AND%20%28%22last%2030%20days%22[PDAT]%29&cmd=Search
>
> [3] https://github.com/BytedProtein/ByProt
>
>
>
>
>
> ## Conclusion:
> **In both cases, where we conducted temporal split evaluations based on structure or sequence-structure, we consistently observed that KWDesign outperforms the baselines.** We hope these additional experiments could address reviewer's concern. Also, please let us know if you have any further questions. Look forward to further discussions!

---

> > ### Comment · Reviewer_4EsM · 2023-11-28
> >
> > Thank you so much for devoting so much time to this topic. I'm impressed by how professional you have been.
> >
> > I have updated the numeric score of my review and have updated the text of my review to reflect my latest thoughts.

---

> > > ### Author Response · Authors · 2023-11-30
> > > **Thanks for your professional review!**
> > >
> > > Thanks for your valuable feedback! We have revised the manuscript accordingly. On page 3 and page 4, we explitly describe the stagewise training procedure and point out the computational reasons behind it. We also appreciate your kind suggestion about using gradient checkpoints, which we will try in the future.

---

### Author Response · Authors · 2023-11-28
**Looking for further discussion regarding the rebuttal**

Dear Reviewers,

We hope this message finds you well. We are writing to kindly follow up on the discussion process of our manuscript. We understand the demands of your time and truly appreciate your effort and expertise in this phase.

However, we would like to bring to your attention that, as there are only two reviewers for the manuscript, the deadline for the rebuttal has been extended to December 1st. This extension provides ample opportunities for further discussion and feedback.

Your insights are crucial for us to enhance the quality of our work. We are eagerly looking forward to your comments and suggestions regarding our rebuttal and supplementary experiments newly added.

Thank you once again for your time and effort in the rebuttal-discussion phase.

Warm regards,

Authors

---

### Meta-Review · Program_Chairs · 2024-01-15

**Metareview:**

The paper studies inverse protein design, i.e., the generation of a protein that folds into a particular shape. The approach, KW-design, is based on the refinement of sequences generated via previous methods. The method brings together several components, including the use of pretrained sequence and structure models to extract multi-modal features, a confidence-aware gated layer to fuse these features, and a memory retrieval mechanism to improve training efficiency. The method is shown to lead to substantial improvement over the current state-of-the-art.

The reviewers liked the problem statement and found the technical approach and the experimental results to be strong. Some aspects of the presentation were hard to follow, and there were also concerns about data leakage. However, these concerns were mostly addressed during the discussion process. Given this, I am enthusiastically recommending acceptance. Please make sure to incorporate the reviewers' feedback into the final version.

**Justification For Why Not Higher Score:**

The paper, while novel and sound, is not exactly spectacular in its originality and quality of results.

**Justification For Why Not Lower Score:**

The methodology is sound, and the results are solid.

---

### Decision · Program_Chairs · 2024-01-16

Accept (poster)